



# Gravity wave instability structures and turbulence from more than one and a half years of OH* airglow imager observations in Slovenia

René Sedlak[1], Patrick Hannawald[1,2], Carsten Schmidt[2], Sabine Wüst[2], Michael Bittner[1,2], and Samo Stanič[3]

[1]Institute of Physics, University of Augsburg, Augsburg, Germany
[2]German Remote Sensing Data Center, German Aerospace Center, Oberpfaffenhofen, Germany
[3]Center for Astrophysics and Cosmology, University of Nova Gorica, Ajdovščina, Slovenia

*Correspondence to*: René Sedlak (rene.sedlak@physik.uni-augsburg.de)

**Abstract.** We analysed 286 nights of data from the OH* airglow imager FAIM 3 (Fast Airglow IMager) acquired at Otlica Observatory (45.93 °N, 13.91 °E), Slovenia between 26 October 2017 and 6 June 2019. Measurements have been performed with a spatial resolution of 24 m pixel$^{-1}$ and a temporal resolution of 2.8 s.

A two-dimensional Fast Fourier transform is applied to the image data to derive horizontal wavelengths between 48 m and 4.5 km in the upper mesosphere / lower thermosphere (UMLT) region. In contrast to the statistics of larger scale gravity waves (horizontal wavelength up to ca. 50 km) we find a more isotropic distribution of directions of propagation, pointing to the presence of wave structures created above the stratospheric wind fields. A weak seasonal tendency of a majority of waves propagating eastward (westward) during winter (summer) may be due to secondary gravity waves originating from breaking primary waves in the stratosphere. We also observe an increased southward propagation during summer, which we interpret as an enhanced contribution of secondary gravity waves created as a consequence of primary wave filtering by the meridional mesospheric circulation.

Furthermore, observations of turbulent vortices allowed the estimation of eddy diffusion coefficients in the UMLT from image sequences in 45 cases. Values range around $10^3 - 10^4$ m$^2$s$^{-1}$ and mostly agree with literature. Turbulently dissipated energy is derived taking into account values of the Brunt-Väisälä frequency based on TIMED-SABER (Thermosphere Ionosphere Mesosphere Energetics Dynamics, Sounding of the Atmosphere using Broadband Emission Radiometry) measurements. Energy dissipation rates range between 0.63 W kg$^{-1}$ and 14.21 W kg$^{-1}$ leading to an approximated maximum heating of 0.2 - 6.3 K per turbulence event. These are in the same range as the daily chemical heating rates, which apparently stresses the importance of dynamical energy conversion in the UMLT.



## 1 Introduction

Fully understanding the contribution of gravity waves to atmospheric dynamics is still a major issue when establishing climate

models. Due to the various sources and mechanisms of interactions the effects of gravity waves have to be represented in these

models using advanced parameterizations (Lindzen, 1981; Holton, 1983; de la Cámara et al., 2016) to cover as many aspects

as it is possible given the restricted model resolution. Gravity waves exist on a large span of time scales ranging from several

hours down to the Brunt-Väisälä (BV) period, which corresponds to ca. 4 - 5 min in the upper mesosphere / lower thermosphere

(UMLT) region (Wüst et al., 2017b) and represents the smallest possible period of gravity waves. They show diverse behaviour

depending strongly on wave properties like their periodicity (Fritts & Alexander, 2003; Beldon & Mitchell, 2009; Hoffmann

et al., 2010; Wüst et al., 2016; Sedlak et al., 2020), which makes it even harder to fully account for them by means of

parameterization. Furthermore, gravity wave generation is not restricted to the ground but can also take place at higher

altitudes, such as secondary wave excitation due to breaking gravity waves (see e.g. Holton & Alexander, 1999; Satomura &

Sato, 1999; Vadas & Fritts, 2001; Becker & Vadas, 2018).

As Fritts & Alexander (2003) state, it is necessary to metrologically capture all parts of the gravity wave spectrum. This

includes especially dynamics on short scales where gravity wave breaking is induced by the development of instabilities. One

of the most prominent features in this context is the formation of Kelvin-Helmholtz instability (KHI), which occurs as a

consequence of a dynamically instable atmosphere due to wind shear (Browning, 1971). Gravity wave instability can also be

of convective nature when growing wave amplitudes lead to a superadiabatic lapse rate (Fritts & Alexander, 2003). In general,

atmospheric instabilities like KHIs often manifest as so-called ripples – periodic structures with small spatial dimensions and

short lifetimes (Peterson, 1979; Adams et al., 1988; Taylor & Hapgood, 1990; Li et al., 2017).

Gravity wave breaking and the conversion of the transported energy into heat takes place in the course of turbulence. Once a

wave breaks and motion shifts from laminar to turbulent flow energy is cascaded to smaller and smaller structures until

viscosity becomes dominant over inertia and energy is dissipated into the atmosphere by viscous damping (see e.g. Lübken et

al., 1987).

The process of turbulence manifests as formation of vortices, so-called eddies. They cause turbulent mixing of the medium,

which is described by the eddy diffusion coefficient $K$. $K$ can be calculated from the eddy radius $r_e$ and the circumferential

velocity $v_e$ by

$$K = \frac{2}{3\pi} r_e v_e \qquad (1)$$

(see e.g. Prölss, 2001. The derivation is outlined in detail in appendix A.). According to Weinstock (1978), knowing $K$ and the

BV frequency $N$, an estimate for the energy dissipation rate $\epsilon$ - the rate at which turbulent kinetic energy is dissipated into heat

at the short-scale end of the energy cascade of the inertial subrange (Li et al., 2016) - can be calculated using

$$K \approx 0.81 \cdot \left(\frac{\epsilon}{N^2}\right). \qquad (2)$$

Gravity wave dissipation predominantly occurs in the upper mesosphere / lower thermosphere (UMLT) region (Gardner et al.,

2002). Hocking (1985) states that the turbulent regime at this altitude manifests on scales shorter than 1 km, which sets high





requirements for measurement techniques at these heights. This is why turbulence investigations in the UMLT are challenging and there are only few values of $K$ available at UMLT heights. Lübken et al. (1997) use rocket measurements to retrieve $K$ and $\epsilon$ in the height range 65 - 120 km. Liu (2009) presents a method for the estimation of $K$ from gravity wave momentum fluxes derived from lidar data. Baumgarten & Fritts (2014) use imaging techniques of mesospheric noctilucent clouds to
investigate the formation of KHIs and the onset of turbulence.

At the same height, remote sensing measurements of the OH* airglow are an established access to UMLT dynamics. The OH* airglow is a layer at an average altitude of ca. 86–87 km with a full width at half maximum (FWHM) of ca. 8 km (Baker & Stair, 1988; Liu & Shepherd, 2006; Wüst et al., 2017b). These include spectroscopic measurements of strong emission lines and the analysis of temperature time series derived from these (Hines & Tarasick, 1987; Mulligan et al., 1995; Bittner et al.,
2000; Reisin & Scheer, 2001; Espy & Stegman, 2002; Espy et al., 2003; French & Burns, 2004; Offermann, 2009; Schmidt et al., 2013, 2018, Wachter et al., 2015; Silber et al., 2016, Wüst et al., 2016, 2017a, 2018), but also two-dimensional imaging in the short-wave infrared (SWIR) range (see e.g. Peterson & Kieffaber, 1973; Hecht et al., 1997; Taylor, 1997; Moreels et al., 2008; Li et al., 2011; Pautet et al., 2014; Hannawald et al., 2016, 2019; Sedlak et al., 2016; Wüst et al., 2019 and many more). The technology of OH* imaging has undergone a rapid technical progress over the last few decades. Proceedings in sensor
technology and optics have provided the possibility to observe the signatures of gravity waves that manifest as periodic brightness variations in infrared images of the OH* airglow layer. The observations range from all-sky imaging of large-scale gravity waves (e.g. Taylor et al., 1997; Smith et al., 2009) to high resolution images of smaller gravity waves (Nakamura et al., 1999) and their breaking processes (Hecht et al., 2014; Hannawald et al., 2016). Hannawald et al. (2016) use an airglow imager called FAIM (Fast Airglow IMager) that is well-suited for the observation of small-scale gravity waves with a high
temporal resolution of 0.5 s. Based on three years of continuous night-time observations at two different Alpine locations Hannawald et al. (2019) show statistics of gravity wave propagation for waves with horizontal wavelengths smaller than 50 km based on data of the same kind of instrument.

In 2016 we put into operation another FAIM instrument (FAIM 3) which still has a high temporal resolution of 2.8 s, but also a high spatial resolution of up to 17 m pixel$^{-1}$ (measurements in zenith direction utilizing a 100 mm SWIR objective lens). We
were not only able to observe wave patterns on extraordinary small scales (smallest horizontal wavelength 550 m) but also the formation of a vortex which we interpret as the turbulent breakdown of a wave front (Sedlak et al., 2016).

From October 2017 to June 2019 the instrument observed the area around the Gulf of Trieste from Otlica Observatory, Slovenia (45.93 N, 13.91°E), which is a partner observatory within the context of the Virtual Alpine Observatory (VAO; https://www.vao.bayern.de). This larger data basis includes further observations of small-scale wave features and turbulence
which are investigated here.

The focus of this paper is on analysing small-scale dynamics in the UMLT region in FAIM 3 images with regard to two aspects:

1.  A statistical analysis of wave parameters on scales below 4.5 km using a 2-dimensional Fast Fourier Transform (2d-FFT). Using the same measurement technique and analysis we are able to directly connect to the short-scale end of the investigations performed by Hannawald et al. (2019).



2.   The estimation of dissipated energy by analyzing multiple episodes of turbulence (such as the one exemplarily presented in Sedlak et al., 2016).

## 2 Instrumentation

FAIM 3 is an OH* airglow imager that has been put into operation in February 2016 at the German Aerospace Center (DLR)
in Oberpfaffenhofen, Germany. It consists of the SWIR camera CHEETAH CL manufactured by Xenics nv, which has a thermodynamically cooled $640 \times 512$ pixels InGaAs sensor array (pixel size $20\,\mu m \times 20\,\mu m$, operating temperature $233\,K$). The camera is sensitive to electro-magnetic radiation in the wavelength range from 0.9 to $1.7\,\mu m$ (for further technical details see Sedlak et al., 2016).

From 26 October 2017 to 6 June 2019 automatic measurements with focus on the OH* airglow emissions have been performed
at Otlica Observatory (OTL) ($45.93\,N$, $13.91\,°E$), Slovenia. FAIM 3 was aligned at a zenith angle of $35°$ and an azimuthal direction of $240°$ (facing approximately into WSW direction). Measurements are only possible during night-time because OH* emissions are not detectable in the presence of the much stronger solar radiation. A baffle was attached to prevent the images from being disturbed by reflections from the lab interior e.g. by moon light. As in Sedlak et al. (2016) the camera was equipped with a $100\,mm$ SWIR lens by Edmund Optics® with aperture angles of $7.3°$ and $5.9°$ in horizontal and vertical direction.
Neglecting the curvature of the Earth, this configuration leads to a trapezium-shaped field of view (FOV) with a size of ca. $182\,km^2$ ($13.1 - 14.1\,km \times 13.4\,km$) at the mean peak emission height of the OH* layer at ca. $87\,km$. The mean spatial resolution is therefore $24\,m\,pixel^{-1}$. Due to the abovementioned measurement geometry the FOV is located above the Gulf of Trieste. The integration time of FAIM 3 is $2.8\,s$, which leads, depending on the season, to the acquisition of ca. 10,000 to 18,000 images per night.

## 3 Data Basis

All in all, in 477 nights image data were acquired by FAIM 3 at OTL. Since OH* airglow observations are only possible under clear sky conditions, cloudy episodes are sorted out by analyzing keograms. This yields 410 clear sky episodes (durations between $20\,min$ and $13\,h$) that are distributed over 286 measurement nights. Thus, ca. 60% of the acquired nights at OTL include suitable OH* observations.
Before being analysed, the images undergo the same preprocessing steps as in Hannawald et al. (2016, 2019) and Sedlak et al. (2016): a flat-field correction is performed and the images are transformed to an equidistant grid, which corresponds to a trapezium-shaped FOV due to the inclination from zenith. For each episode the average image is subtracted to ensure that all remnants of fixed patterns are removed (e.g., reflections of the objective lens in the laboratory window during bright nights). Due to the small FOV of FAIM 3 we renounce the application of a star removal algorithm to avoid an interpolation of too



many pixels. In order to extract periodic signatures a two-dimensional Fast Fourier Transform (2d-FFT) is applied to squared cut-outs of each image, so that neither dimension is favored by the analysis. These cut-outs were chosen to have a side length of 406 pixels (equals ca. 9.7 km) as this is the largest possible square fitting into the transformed images. The 2d-FFT is performed on the squared image cut-out as described by Hannawald et al. (2019). A fitted linear intensity gradient is subtracted and a Hann window is applied to reduce leakage effects. A local maximum filter is applied to automatically find peaks in the

spectra and thus plane wave structures, which allows identifying and analyzing single wave events. Zero-padding on the images is used to improve this identification of peaks in the spectra. Hannawald et al. (2019) present a statistical analysis of gravity waves with horizontal wavelengths between 2 and 62 km (with focus on waves with horizontal wavelengths larger than 15 km). With FAIM 3 having a smaller FOV and a higher spatial resolution than the FAIM instrument used therein, we are now able to present statistics of gravity wave parameters that tie in almost seamlessly with the statistics of longer-scale waves of

Hannawald et al. (2019): due to the spatial resolution and the FOV size we cover the horizontal wavelength range from 48 m to 4.5 km. Wave structures with horizontal wavelengths of half the FOV size still showed a strong bias toward phases 0 or $\pi$. Extensive testing showed that this effect disappeared when lowering the upper wavelength limit to 4.5 km.

Observed wave structures have to meet several quality criteria in order to be considered a wave event. A wave structure has to be present for at least 20 s and has to be found in at least eight images. This is in contrast to Hannawald et al. (2019) who

demand wave signatures to be present for at least 120 s and to appear in at least 100 images within this episode, stating that these restrictions specifically filter out many transient and small-scale wave features as they want to focus on larger persistent waves.

Furthermore, FAIM 3 wave events are considered if they have an amplitude of at least 25 % of the maximum observed wave amplitude. Wave structures with this amplitude can just be recognized in the image by the eye. Demanding all the quality

criteria mentioned above a total number of 5697 wave events remains. Further restricting these criteria has not significantly altered the distributions of the wave parameters that are presented in the following.

While the identification of wave structures is done automatically by the 2d-FFT, finding turbulent vortices is done by hand. Turbulent eddy formation can be well recognized by eye when viewing the episodes in the dynamical course of a video sequence. However, the combined effect of these vortices having a certain variety of shapes and sizes, being almost invisible

in single images without comparison to preceding or successive images, and causing (compared to other features such as wave fronts) rather small brightness fluctuations in the images hampers strongly the application of image recognition algorithms. For the given data basis 45 episodes of turbulence with sufficient quality to derive vortex parameters are found. The dates along with the respective turbulence parameters are summarized in Table 1.

For both, gravity wave statistics (section 4.1) and the calculation of the energy dissipation rate (section 4.2) the BV frequency

is required, which is adapted from the climatology presented by Wüst et al. (2020). It is based on TIMED-SABER (Thermosphere Ionosphere Mesosphere Energetics Dynamics, Sounding of the Atmosphere using Broadband Emission Radiometry) temperature data and takes into account the seasonal variability of the angular BV frequency. The climatology of



the grid point (45° N, 10° E) is used, which is closest to our FOV. Depending on the day of the year (DoY) the BV frequency is given by

$$N = 2.20 \cdot 10^{-2}\text{s}^{-2} + 0.19 \cdot 10^{-2}\text{s}^{-2} \sin\left(\frac{2\pi}{324.51\text{d}} \cdot DoY - 2.02\right) + 0.05 \cdot 10^{-2}\text{s}^{-2} \sin\left(\frac{2\pi}{180.00\text{d}} \cdot DoY + 1.61\right). \quad (3)$$

We use an uncertainty of $\pm 5\,\%$ as according to Wüst et al. (2020) 91 % of their data lie within this range around the harmonic approximation. The BV period is then referred to as $\tau_{BV} = \frac{2\pi}{N}$.

## 4 Results

### 4.1 Statistics of Wave Parameters

The wave statistics are presented in Figure 1. For each wave event the individual BV period is calculated based on Eq. (3) and the DoY. Ca. 63 % of the observed wave events have a period longer than the respective BV period and will be referred to as gravity wave events in the following. Their statistical contribution is highlighted in grey in Figure 1.

Wave periods range from 21 s to 1498 s (25 min). The median wave period is 359 s (6 min). The median value of gravity wave periods is found at 517 s (8.6 min). The maximum phase speed is 17.6 m s$^{-1}$ with an average value of 7.9 m s$^{-1}$ and a standard

deviation of 3.3 m s$^{-1}$. As concerns the zonal distribution, 50.7 % (49.3 %) of the gravity wave events have an eastward (westward) component (consequently no gravity waves with zonal phase speed zero have been observed) and the mean velocity in eastward (westward) direction is 5.4 m s$^{-1}$ (5.3 m s$^{-1}$) with a standard deviation of 3.0 m s$^{-1}$ (3.1 m s$^{-1}$). We find a small seasonal effect in the distribution of zonal phase speeds: 53.8 % of the waves have an eastward component and 46.2 % a westward component when only considering the winter months December to February while during summer from June to August 47.9 %

of the waves have an eastward component and 52.1 % a westward component. In meridional direction 44.4 % (53.5 %) of the gravity wave events have a northward (southward) component and the mean meridional phase speed is 4.7 m s$^{-1}$ (5.3 m s$^{-1}$) in northern (southern) direction with a standard deviation of 3.0 m s$^{-1}$ (3.2 m s$^{-1}$). Events with meridional phase speed zero have not been considered for the mean values.

### 4.2 Wave Dissipation

The eddy parameters needed for the calculation of the eddy diffusion coefficient (Eq. 1) are determined manually from the image series of the 45 observations of turbulence. It has to be kept in mind that we are deriving properties of a three-dimensional movement from two-dimensional data.

We assume the vortices to rotate in a perfect circular shape. The lateral expansion creates the impression of a rotating cylinder. Coherently moving structures give indication of the horizontal velocity vector. Unless the rotational axis is aligned

perpendicular to the image plane, the three-dimensional vortex rotation manifests as more than one coherent structure that is moving against or overtaking each other (see e.g. Sedlak et al., 2016; Figure 6 therein). During data inspection we noticed that the orientation of the rotational axis can be aligned in any direction. It tends to be parallel to the image plane when it evolves



directly from the crests of a breaking wave. However, we could also observe eddies rotating around an axis aligned almost perpendicular to the image plane. An example of a rotating vortex within a FAIM 3 snapshot on 4 November 2018 at

19:36:41 UTC is displayed in Figure 3a. The rotational axis and the direction of rotation are marked therein and on an actual cylinder (Figure 3b) for clarification. Since it is very difficult to identify a vortex structure in a single picture we have attached a video sequence of this episode (Video 1). The vortex radius and velocity are read from the images. Besides measuring the vortex rotation, it has also to be taken care of the overall image: if additional to the eddy movement all structures in the FOV are moving into a common direction, this background motion has to be subtracted. In the example shown above the vortex is

advected toward the left corner. The distance between camera and observed vortex is much larger than the expansion of the vortex along the rotation axis so that falsifications arising from different perspectives of the vortices can be neglected. This principle is illustrated in Figure 4. As the vortices are three-dimensional the alignment of the rotational axis should not affect the value of the vortex parameters in the images: it does not matter if the axis is aligned perpendicular, parallel or in any other angle to the image plane, the vortex size will be accessible from the two-dimensional projection of the image assuming circular

eddy movement. The same holds for the circumferential velocity since both the radius and the circulation time remain unchanged. However, perfectly circular eddy rotation does not necessarily occur in nature. Deviations from circularity can lead to both over- and underestimation of vortex sizes depending on the vortex orientation. Since isotropy is one of the characteristic properties of turbulent movements one may presume that from a statistical point of view both cases occur equally so that no systematic error is made.

As stated in section 3 we found 45 episodes of turbulence that allowed the derivation of the vortex radius and circumferential velocity. The resulting eddy diffusion coefficients $K$ are shown in Figure 5. We assume a general read-out error of $\pm 3$ pixels, which corresponds to a distance of $\pm 72\,\mathrm{m}$. The circumferential velocity is determined by reading the distance a patch on the rotating cylinder surface covers within an episode of at least ten images, which corresponds to a time span of $28\,\mathrm{s}$. Thus, the circumferential velocity is estimated with an error of $\pm 2.6\,\mathrm{m\,s^{-1}}$. The arising uncertainties of $K$ are calculated following the

rules of error propagation.

The values of $K$ range from $0.12$ to $1.94 \cdot 10^4\,\mathrm{m^2 s^{-1}}$. The mean value is $0.76 \cdot 10^4\,\mathrm{m^2 s^{-1}}$ (standard deviation of $0.53 \cdot 10^4\,\mathrm{m^2 s^{-1}}$) and we retrieve a median of $0.59 \cdot 10^4\,\mathrm{m^2 s^{-1}}$. It is difficult to exactly quantify the error of manual parameter determination from the images. However, in this work we rather focus on the order of magnitude of $K$. When calculating $K$ two distance values are read from the images (one for the vortex size and one for the determination of the circumferential

speed). Considering Eq. (1), a mistake of factor 10 is made for $K$ if these distances are misread by a factor of at least $\sqrt{10}$. The shortest (and therefore most difficult to determine) diameter in our analysed examples was $768\,\mathrm{m}$. For an error of one order of magnitude of $K$ this distance must be misread as either shorter than $243\,\mathrm{m}$ or longer than $2428\,\mathrm{m}$, i.e. a distance of 32 pixels must be wrongly interpreted as shorter than 9 pixels or longer than 101 pixels. This lies far beyond the read-out uncertainty of $\pm 3$ pixels we introduced above and can be assumed to be much worse than any read-out error one would normally make.





The energy dissipation rate $\epsilon$ can be estimated from the eddy diffusion coefficient $K$ according to Eq. (2) using the BV frequency as described by Eq. (3).

As can be seen in Figure 5 the energy dissipation rate of the observed turbulence events is in the range $0.63 - 14.21\,\mathrm{W\,kg^{-1}}$. Assuming the duration of dissipation being equal to the lifetime of the vortex the energy dissipation rate can be converted into the amount of dissipated energy per mass. This is only done for those vortices that both form and decay within the FOV. The

time intervals of dissipation are between $146\,\mathrm{s}$ and $922\,\mathrm{s}$ $(2.4 - 15.4\,\mathrm{min})$ and can also be found in Table 1. Events are labelled as 'out of FOV' or as 'clouds' if either the formation or the decay of the vortex cannot be observed. No further analysis is performed for these events.

Multiplying energy dissipation rate and duration of dissipation equals the energy per mass that is released in the turbulent process. We retrieve values between 220 and $6346\,\mathrm{J\,kg^{-1}}$. Given that the released energy is entirely converted into heat we can

make a rough estimate of the resulting temperature change by assuming isobaric conditions (may be approximately fulfilled due to the stable stratification of the atmosphere and small vertical dimension of eddies) and dividing energy per mass by the specific heat capacity of dry air ($10^3\,\mathrm{J\,K^{-1}\,kg^{-1}}$). The resulting temperature changes in this work are in the range $0.2 - 6.3\,\mathrm{K}$. $64\,\%$ (21 out of 33) of these values are larger than one Kelvin. All values of dissipated energy per mass and maximum temperature change can be found in Table 1.

Since we now have a time series of eddy diffusion coefficients we can compare them to gravity wave activity in the UMLT above OTL. Parallel to FAIM 3, SWIR spectrometers called GRIPS (GRound-based Infrared P-branch Spectrometer) instruments deliver time series of OH* rotational temperatures derived from the OH(3-1) P-branch ($1.5\,\mu\mathrm{m}$-$1.6\,\mu\mathrm{m}$) at an initial temporal resolution of $15\,\mathrm{s}$. Unlike the general instrument details discussed by Schmidt et al. (2013), the GRIPS 9 at OTL has a reduced aperture angle of $6.2$ FWHM increasing its responsivity to smaller structures. As described in Sedlak et al. (2020),

gravity wave activity – the so-called significant wavelet intensity (SWI) – for the periods $6 - 480\,\mathrm{min}$ (period resolution $1\,\mathrm{min}$) can be calculated by applying a wavelet analysis to these temperature time series. The FOV of GRIPS 9 is also located above the Gulf of Trieste and in ca. $30\,\mathrm{km}$ distance to the FAIM 3 FOV and has a size of approximately $13\,\mathrm{km}\,\mathrm{x}\,19\,\mathrm{km}$. Since the spectroscopic observations are averaged over the entire FOV, GRIPS is most sensitive for gravity waves with horizontal wavelengths of several hundreds of kilometres (Wüst et al., 2016). The time series of nocturnal SWI is restricted to those

nights that exhibited at least one of the turbulence episodes presented above and the correlation between the SWI and the eddy diffusion coefficient has been calculated. If there are observations of more than one vortex during one night, the respective eddy diffusion coefficients are averaged to their mean value. The Pearson correlation coefficient and the P value (significance test) are presented in Figure 7. We find a slight but significant anticorrelation for gravity wave periods in the range $122 - 207\,\mathrm{min}$. For these periods the mean value of the correlation coefficient is $-0.46$. The highest coefficient of anticorrelation

is $-0.52$ at a period of $178\,\mathrm{min}$.



## 5 Discussion

The directions of propagation are quite uniformly distributed over all quadrants as can be seen in Figure 2. The strong tendency to north-eastern direction in summer and to the (south-)west in winter as observed by Hannawald et al. (2019) for medium-scale gravity waves cannot be confirmed for the waves observed here. Only the north-western component these authors observed during winter at Mt. Sonnblick in Austria with the FOV being positioned north of the Alps also appears in our data during autumn, winter and spring. During summer we find a conspicuous majority of waves propagating into southern direction.

The number of waves propagating eastward and westward is almost equal for the entire data set. However, as stated in section 4.1, more waves are oriented in eastward direction during winter (positive zonal phase speed) and in westward direction during summer. Although this tendency is quite weak, it contradicts the distribution that is expected for gravity waves being created in the lower atmosphere and propagating upward, being subdued to stratospheric wind filtering. The eastward oriented mean wind profile during winter would lead to mainly westward propagating gravity waves reaching the UMLT without encountering critical levels. The reversed stratospheric winds during summer would consequently allow some more eastward travelling gravity waves to propagate upward (see e.g. Hoffmann et al., 2010; Hannawald et al., 2019). Since the highest observed phase speed of waves with periods longer than the BV period is only $17.6\,\mathrm{m\,s^{-1}}$, it can be assumed that in the majority we do not observe gravity waves that are originating from low altitudes and are fast enough not to be blocked by the stratospheric wind fields.

Considering the directional distribution, it is possible that the major part of our waves may be secondary gravity waves (see e.g. Becker & Vadas, 2018), being generated at greater heights by breaking gravity waves. Secondary gravity waves can either have larger wavelengths and phase speeds than the primary wave if they are created by localized momentum deposition (Vadas & Becker, 2018) or smaller wavelengths and phase speeds if they are induced by the nonlinear flow (wave-mean flow and wave-wave interactions; see e.g. Bacmeister & Schoeberl, 1989; Franke & Robinson, 1999; Bossert et al., 2017). The former type of secondary gravity waves exhibits a rather broad spectrum of wave parameters with horizontal wavelengths longer than $500\,\mathrm{km}$ and horizontal phase speeds between 50 and $250\,\mathrm{m\,s^{-1}}$ (Vadas et al., 2018), resulting in periods longer than ca. $30\,\mathrm{min}$. The wave structures found in this work have smaller horizontal wavelengths, phase speeds and periods and could therefore be more likely related to the latter type of secondary waves created by nonlinearities. However, these small-scale secondary waves are unlikely to propagate large vertical distances due to their small horizontal phase speeds (Becker & Vadas, 2018). They have to be generated at even higher altitudes, i.e. close to the mesopause, to be observable with OH* airglow imagers. Hannawald et al. (2019) e.g. deduce from their observations that not only the zonal stratospheric winds but also the meridional circulation in the mesosphere might play a vital role in filtering gravity waves. The meridional mesospheric circulation is oriented southward during summer and northward during winter, being much stronger during summer with ca. $10 - 14\,\mathrm{m/s}$ (Yuan et al., 2008). Simulations by Becker & Vadas (2018) show that advection by the background wind determines the direction of a newly created secondary wave. Based on these aspects, the accumulation of southward oriented waves we





observe during summer could be a hint for gravity waves being filtered by the mesospheric circulation and generating
subsequent secondary waves with shorter wavelengths and periods, that are provided with a southward phase speed due to
advection. This theory is also in good agreement with our observed meridional phase speeds: in the abovementioned velocity
range of the summerly meridional mesospheric circulation (10‑14 m/s) meridional phase speeds are southward in 62 % of
cases.

The small spatial scales of the wave structures we observe are also typical for ripple structures as they were already observed
with FAIM 3 (Sedlak et al., 2016). Their short life spans are not excluded by our quality criteria. Tuan et al. (1979) state that
oscillations of this type are usually excited at periods of 4‑10 min, which would explain the large number of wave events we
observe in this period range. Observing ripple structures, it would not be surprising to obtain a certain diversity of directions
of propagation. In principal, ripples originating from convective instabilities tend to be aligned perpendicular to the wave fronts
of the initial wave, whereas ripples arising from dynamic instabilities form parallel to the initial wave fronts (Andreassen et
al., 1994; Fritts et al., 1997; Hecht et al., 2000). However, it has been reported that ripples can be rotated by the background
wind and that ripples may even be created by a combination of both dynamical and convective instability (Fritts et al., 1996;
Hecht, 2004).

Li et al. (2017) report that ripples are hard to distinguish from small-scale gravity waves. Height-resolved measurements of
the horizontal wind would be needed to determine the local wind shear and make a profound statement about atmospheric
instability. Nevertheless, capturing structures related to instability is not unlikely, considering the numerous observations of
turbulent vortices with the FAIM 3 setup.

There are still very few measurements of turbulent eddy diffusion coefficients in the UMLT. Lübken (1997) reports $K$ to be
around $10^1 - 10^2$ m$^2$s$^{-1}$ at a height of 87 km at high latitudes. Hodges (1969) states that the eddy diffusion coefficient caused
by gravity waves is typically around $10^3$ m$^2$s$^{-1}$. According to the CIRA (Committee on Space Research (COSPAR)
International Reference Atmosphere) climatology of 1986 (NASA National Space Science Data Center, 2007) global values
range between magnitudes of $10^2$ and $10^3$ m$^2$s$^{-1}$. LIDAR measurements above New Mexico, USA deliver values that vary
strongly around a magnitude of $10^2$ m$^2$s$^{-1}$ (Liu, 2009). Smith (2012) notes that the WACCM (Whole Atmosphere Community
Climate Model) climatology exhibits rather small values with magnitude $10^1$ m$^2$s$^{-1}$ and that the huge discrepancies of $K$
estimates cannot be fully explained yet. The here-presented values of $K$ exhibit a magnitude of $10^3 - 10^4$ m$^2$s$^{-1}$, which partly
agrees with recent results, although some of our values are higher. The vortices we observe do not necessarily mark the small-
scale end of the energy cascade. It could be possible that the energy is cascaded further to a larger number of smaller eddies
that are no longer visible to our instrument. Parallel in-situ measurements (e.g. lidar, rockets) could be used to estimate the
significance of this effect. Additionally, it has to be kept in mind that the values compared here arise from different
measurement techniques with different horizontal, vertical and temporal resolutions, so that the accessible scales are not
necessarily identical due to the observational filter effect. Nevertheless, the agreement of the above-mentioned authors on eddy
parameters in the UMLT is quite good, considering the fact that energy dissipation rate in the upper troposphere and lower
stratosphere varies by a factor of more than five orders of magnitude (Li et al., 2016).



The derivation of the vortex parameters performed here is challenging due to the blurred shape of dynamic signatures in the OH* layer. Circumferential speed and vortex radius have been extracted manually by measuring distances in the images and

calculating distances from pixel values and we tried to quantify the read-out error by providing a measurement uncertainty and minimize it by repeating the analysis workflow on the same data multiple times. All in all, it seems possible to derive turbulence parameters like the eddy diffusion coefficient and energy dissipation rate from high resolution imager data.

The values of eddy diffusion coefficients derived here show significant anticorrelation with gravity wave activity in the period range 122 - 207 min. One may assume that turbulent vortices we observe could be predominantly attributed to breaking gravity

waves in this period range. If this was the case, stronger wave breaking would manifest as higher eddy diffusion coefficients and result in a lower activity of gravity waves with periods 122 - 207 min in the UMLT. However, especially observations of period-resolved gravity wave activity at altitudes below would be needed to confirm this assumption.

Assuming that the turbulently dissipated energy is entirely converted into heat we find temperature changes of 0.2 - 6.3 K that occur within time spans of 2.4 − 15.4 min. Marsh (2011) report chemical heating rates in the atmosphere to be around 3 − 4 K

per day. Given that our analysed episodes are typical representatives of turbulent wave breaking, dynamical heating by gravity wave dissipation would deliver the same effect within few minutes as does chemical heating during an entire day.

## 6 Summary

We present an analysis of small-scale wave dynamics from OH* imager data acquired between 26 October 2017 and 6 June 2019 at Otlica Observatory, Slovenia. Measurements have been performed with the imager FAIM 3, which has a spatial

resolution of ca. 24 m pixel$^{-1}$ and a temporal resolution of 2.8 s.

Wave structures in the images are systematically identified by applying a 2d-FFT to nocturnal image sequences during clear sky episodes. All wave events meeting our persistency criteria were used to derive a statistical analysis of wave structures with horizontal wavelengths between 48 m and 4.5 km. 63 % of the wave events have a period longer than the BV period and may be tentatively considered as gravity waves. We find an isotropic distribution of directions of propagation, which indicates that

wave structures may be mostly created above the stratospheric wind fields. However, a weak seasonal dependency is found: zonal directions of wave propagation are slightly more eastward during winter and westward during summer. We speculate these to be generated by breaking gravity waves in the course of wind filtering, receiving their zonal direction through advection by the background wind. We find a stronger tendency of southward propagation during summer, which may point to a vital role of gravity wave filtering and excitation of secondary waves by the meridional mesospheric circulation. It is

possible that secondary waves and instability features represent the majority of our observed waves.

Furthermore, we estimated turbulence parameters from 45 episodes of vortex observations. The derived values of eddy diffusion coefficients are in the range around $10^3 − 10^4$ m$^2$s$^{-1}$ and agree mostly with earlier results from rocket and lidar measurements and simulations. Considering the respective values of the BV frequency as calculated by Wüst et al. (2020) we retrieve energy dissipation rates between 0.63 W kg$^{-1}$ and 14.21 W kg$^{-1}$, that cause estimated heatings by 0.2 - 6.3 K per



turbulence event. These have the same order of magnitude as the daily chemical heating rates as reported by Marsh (2011). Given that the observed events are representative of typical processes of gravity wave dissipation, this emphasizes the importance of carefully integrating gravity wave turbulence into climate simulations.

Being able to derive reasonable values of UMLT turbulence parameters from imager data represents an important progress for measurement techniques of atmospheric dynamics. Airglow imagers are much cheaper and more flexible than rockets or lidars.

Considering the huge amount of data, artificial intelligence could be used in the future to identify and analyse turbulent episodes.

**Appendix A**

Derivation of the eddy diffusion coefficient K (Eq. (1))

The derivation of the eddy diffusion coefficient shown here is based on Prölss (2001). It is assumed that a cylindrical vortex with radius $r_e$ and cylinder height $d$ is excited in the course of turbulence, which rotates around an axis perpendicular to the circular cylinder bases (Figure 8). We consider an atmosphere of particle number density $n$ (varying with height $z$) in which a side gas $i$ with particle number density $n_i$ exhibits a concentration gradient $\frac{d\left(\frac{n_i}{n}\right)}{dz}$ (concentration as ratio of particle numbers of side gas and total atmosphere) perpendicular to the rotational axis. Maximum mixing results when the cylinder rotates half a

turn, i.e. when the cylinder segment of high concentration and the cylinder segment of low concentration switch places. This induces a net particle flux through the center plane $A$ of the cylinder.

Let us consider two cylinder segments 1 and 2 of thickness $ds$, width $2b$ and depth $d$ at distance $\pm s$ from the center plane, thus having the volume $dV = d\, 2b\, ds$. The number of side gas particles they contain is

$$(dN_i)_1 = dV\, n_1 \left(\frac{n_i}{n}\right)_1 \tag{4}$$

and

$$(dN_i)_2 = dV\, n_2 \left(\frac{n_i}{n}\right)_2 = dV\, n_2 \left[\left(\frac{n_i}{n}\right)_1 + \frac{d\left(\frac{n_i}{n}\right)}{dz}\, 2s + \cdots\right]. \tag{5}$$

During half a vortex rotation the segments 1 and 2 move towards the center plane. $n$ and $n_i$ change with height while $\frac{n_i}{n}$ remains constant ($n(z) = n$ at the center plane). This results in a net exchange of

$$(dN_i)_A = [(dN_i)_1 - (dN_i)_2]_A \approx -\, dV\, n\, \frac{d\left(\frac{n_i}{n}\right)}{dz}\, 2s \tag{6}$$

side gas particles through the center plane $A = 2\, r_e\, d$ (i.e. we subtracted Eq. (5) from Eq. (4)). With a circumferential velocity $v_e$ and an exchange time of $\Delta t = \frac{\pi r_e}{v_e}$ for half a rotation this leads to a differential particle flux density of

$$d\Phi_e = \frac{(dN_i)_A}{A\,\Delta t} = -dV\, n\, \frac{d\left(\frac{n_i}{n}\right)}{dz}\, 2s\, \frac{1}{A\,\Delta t} = -d\, 2b\, ds\, n\, \frac{d\left(\frac{n_i}{n}\right)}{dz}\, 2s\, \frac{1}{2\, r_e\, d}\, \frac{v_e}{\pi r_e} = -\, 2\sqrt{r_e{}^2 - s^2}\, ds\, n\, \frac{d\left(\frac{n_i}{n}\right)}{dz}\, s\, \frac{1}{r_e}\, \frac{v_e}{\pi r_e}. \tag{7}$$





Please note that $b$ has been replaced by $\sqrt{r_e^2 - s^2}$ according to the Pythagoras Theorem.

Integration over all cylinder segments yields the total flux density

$$\Phi_e = \int_{s=0}^{r_e} d\Phi_e = -\frac{2}{\pi} \frac{v_e}{r_e^2} n \frac{d\left(\frac{n_i}{n}\right)}{dz} \int_{s=0}^{r_e} \sqrt{r_e^2 - s^2}\, s\, ds . \tag{8}$$

The integral can be solved by substituting $x := r_e^2 - s^2$ and therefore $\frac{dx}{ds} = -2s$.

Then

$$\int_{s=0}^{r_e} \sqrt{r_e^2 - s^2}\, s\, ds = -\frac{1}{2} \int_{r_e^2}^{0} \sqrt{x}\, dx = -\frac{1}{3}\left[x^{\frac{3}{2}}\right]_{r_e^2}^{0} = \frac{1}{3} r_e^3 \tag{9}$$

so that

$$\Phi_e = -\frac{2}{3\pi} r_e v_e n \frac{d\left(\frac{n_i}{n}\right)}{dz}. \tag{10}$$

Analogously to Fick's law of molecular diffusion we can write the flux density of eddy diffusion as

$$\Phi_e = -K n \frac{d\left(\frac{n_i}{n}\right)}{dz} \tag{11}$$

where

$$K = \frac{2}{3\pi} r_e v_e$$

is the eddy diffusion coefficient.

**Data availability**

The data are archived at the WDC-RSAT (World Data Center for Remote Sensing of the Atmosphere). The FAIM and GRIPS instruments are part of the Network for the Detection of Mesospheric Change, NDMC (https://ndmc.dlr.de).

**Author contribution**

The conceptualisation of the project, the funding acquisition as well as the administration and supervision were done by MB and SW. The operability of the instrument was assured by RS. SS provided us the opportunity to set up our instrument at Otlica Observatory and took care of the maintenance. The algorithm for retrieving wave statistics was written by PH. The analyses of wave statistics and turbulence from FAIM 3 images as well as the visualization of the results were performed by RS. Setup, operation and data reduction for GRIPS 9 was done by CS. The interpretation of the results benefited from fruitful discussions between PH, CS, SW, MB, and RS. The original draft of the manuscript was written by RS. Careful review of the draft was performed by all co-authors.



## Competing interests

The authors declare that they have no conflict of interest.

## Acknowledgement

This research received funding from the Bavarian State Ministry of the Environment and Consumer Protection by grant number TKP01KPB-70581 (Project VoCaS-ALP).

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



**Figures**

**Figure 1. Statistical distribution of observed parameters for wave events with horizontal wavelengths between 48 m and 4.5 km from 26 October 2017 to 6 June 2019 at Otlica, Slovenia. The contribution of wave events with a period longer than the respective BV period is coloured in grey. a) Period. b) Absolute horizontal phase speed. c) Zonal phase speed. d) Meridional phase speed.**



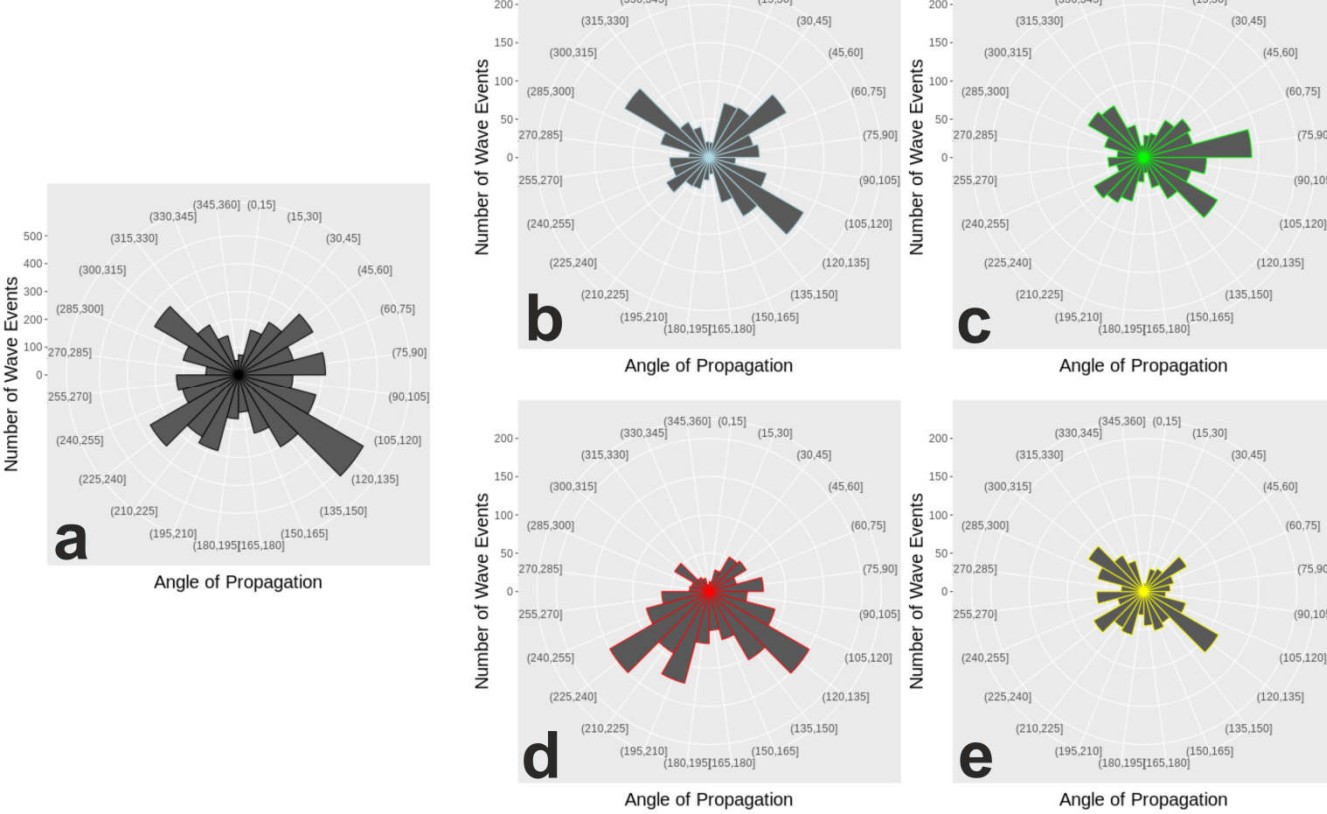

**Figure 2.** Statistical distribution of observed directions of propagation for wave events with horizontal wavelengths between 48 m and 4.5 km from 26 October 2017 to 6 June 2019 at Otlica, Slovenia. a) All. b) Winter (Dec-Jan-Feb). c) Spring (Mar-Apr-May). d) Summer (Jun-Jul-Aug). e) Autumn (Sep-Oct-Nov).





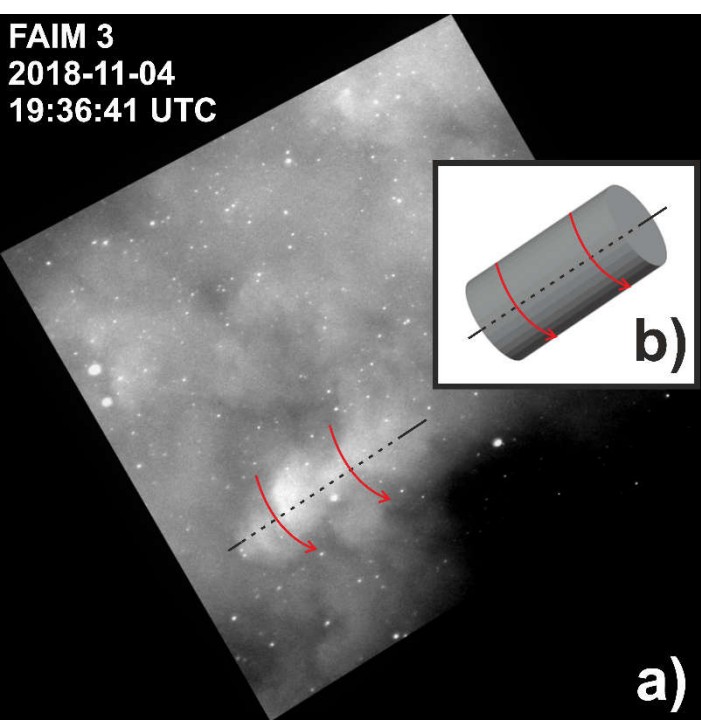

**Figure 3. (a) Snapshot (19:36:41 UTC) of the turbulence episode from 4 November 2018 between 19:32:24 and 19:40:59 UTC. The rotational axis (black line) and movement (red arrows) of a vortex are marked in the picture and on a cylinder (b) to guide the eye.**
**The rotation of the vortex is more apparent in the video supplement to this article (Video 1).**

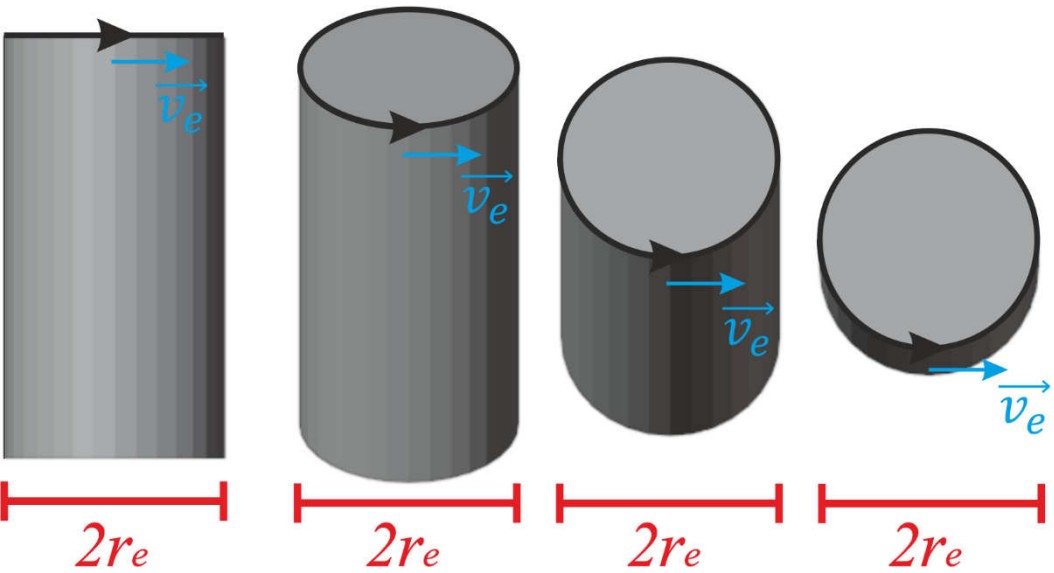

**Figure 4. Schematic view of a rotating cylinder from different angles. In a 2-dimensional projection, the quantities we read from the images – the vortex radius $r_e$ and the circumferential velocity $\overrightarrow{v_e}$ – remain the same for different perspectives.**






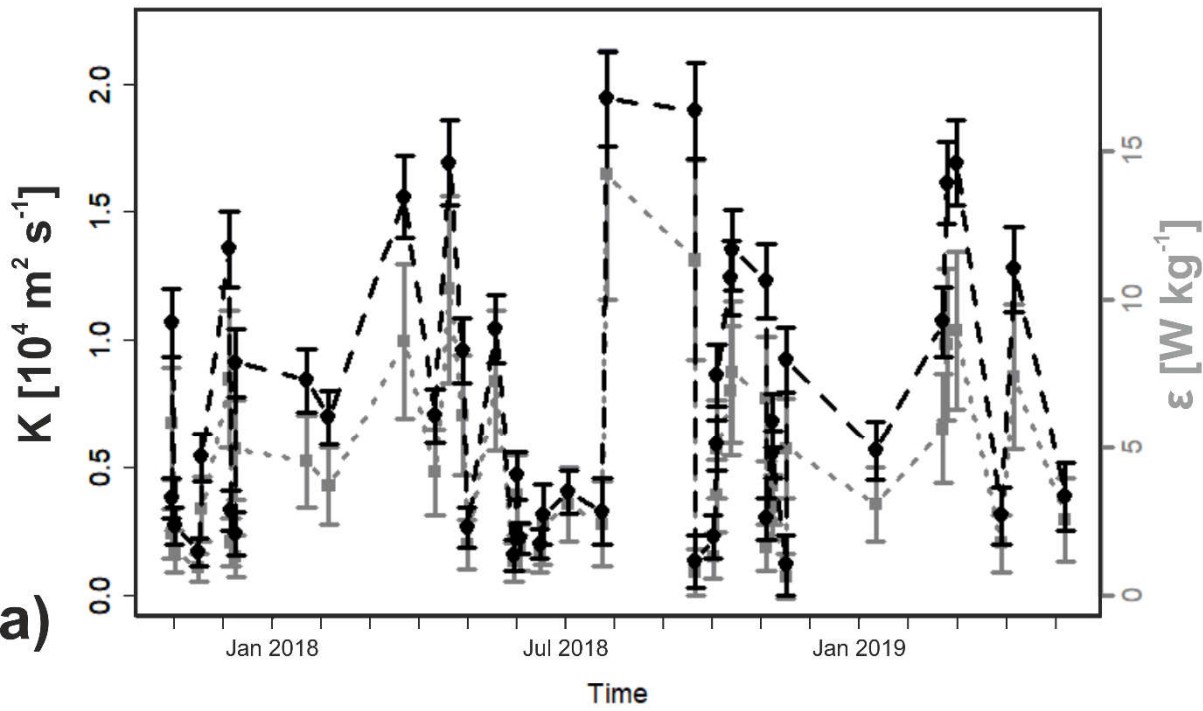

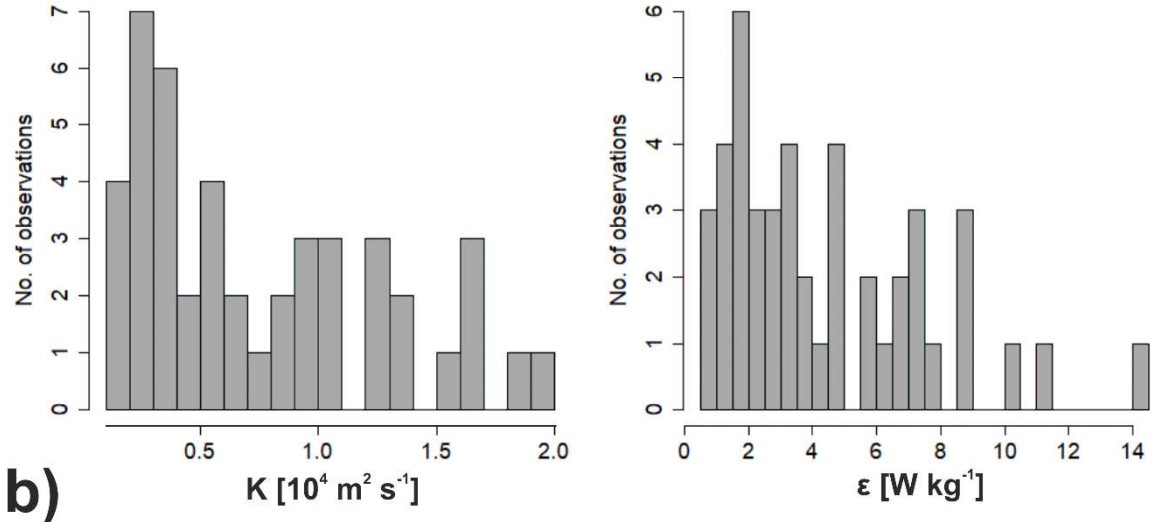

**Figure 5. a) Temporal evolution of eddy diffusion coefficients (black) and energy dissipation rates (grey) of observed turbulence events at OTL (see Table 1). b) Histograms of K and ε.**



**Figure 6. a)** Temporal evolution of dissipated energy per mass [J/kg] and respective temperature change [K] of observed turbulence events at OTL (see Table 1), assuming isobaric conditions and complete conversion of dissipated energy into heat. **b)** Histogram of dissipated energy per mass and temperature change.




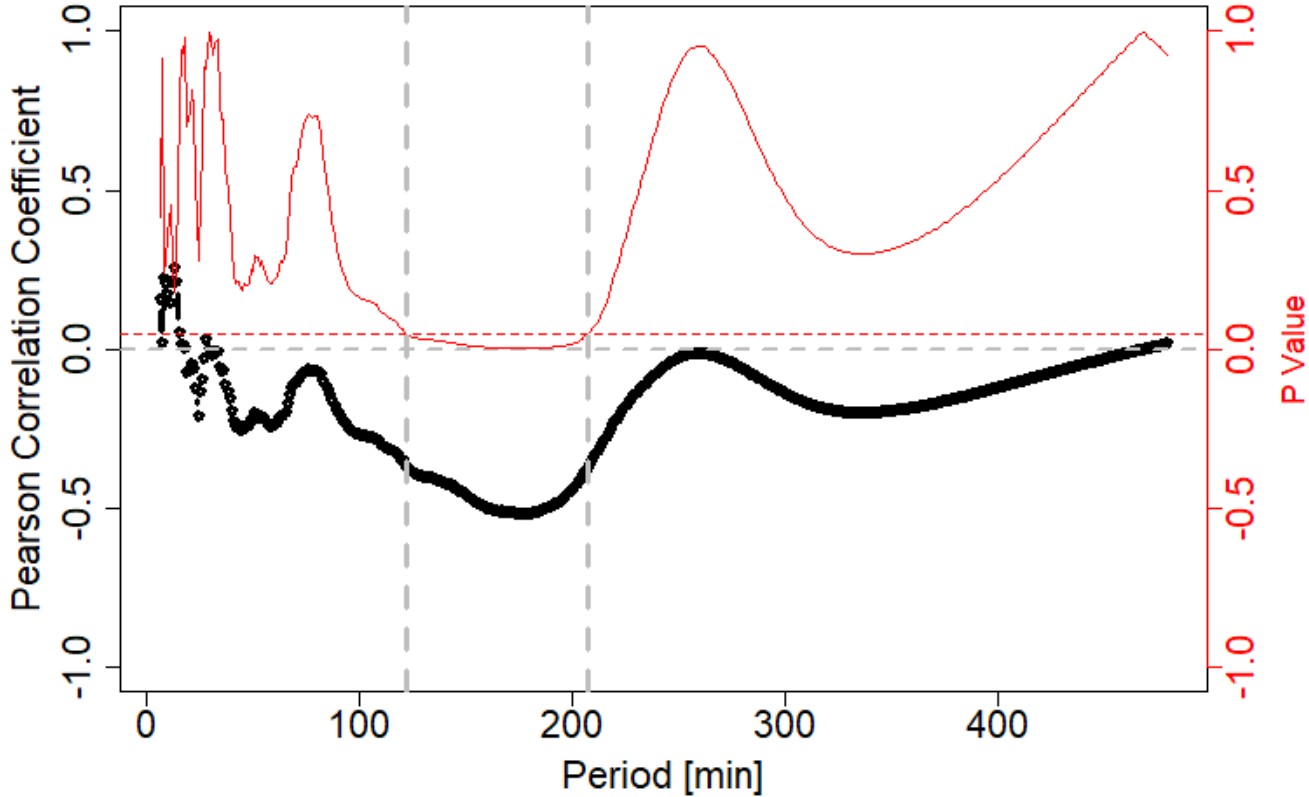

**Figure 7. Pearson correlation coefficient (black) between gravity wave activity (SWI) from GRIPS data and eddy diffusion coefficients from FAIM 3 data above OTL. The P value is plotted in red. For all P values of 0.05 (red horizontal line) or equal the correlation coefficient is considered significant. A significant anticorrelation between SWI and the eddy diffusion coefficient is found between periods of 122 min and 207 min (grey vertical lines).**




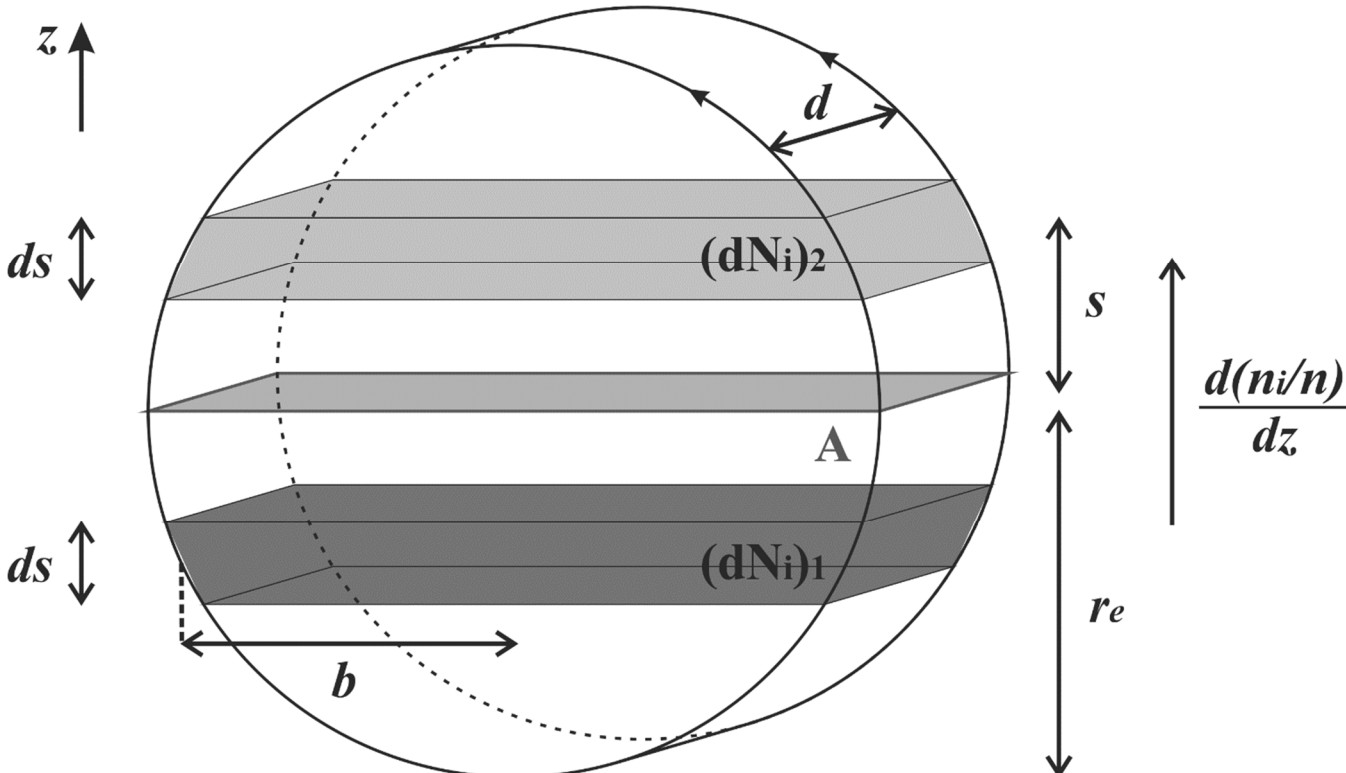

**Figure 8. Concept of cylindrical mixing according to Prölss (2001). Description in text.**



**Tables**

**Table 1.** Episodes of turbulence observed at OTL and derived parameters from the image sequences. The BV frequency is according to the climatology based on TIMED-SABER measurements as presented by Wüst et al. (2020). The duration of the turbulence events could not be determined if the vortex was not visible during its entire life span due to being partly outside the FOV ('out of FOV') of FAIM 3 or covered by clouds ('clouds'). In these cases, we noted the dissipated energy per mass and the maximum temperature change as 'not available' (NA).

| Date | DoY [d] | K [$10^4$ m²/s] | Duration [s] | Angular BV frequency [$10^{-2}$ 1/s] | $\epsilon$ [W/kg] | Diss. energy per mass [J/kg] | Max. temperature change [K] |
|---|---|---|---|---|---|---|---|
| 2017-10-30 | 303 | 0.38 ± 0.08 | out of FOV | 2.10 ± 0.11 | 2.07 ± 0.63 | NA | NA |
| 2017-10-30 | 303 | 1.07 ± 0.13 | 244 | 2.10 ± 0.11 | 5.82 ± 1.31 | 1420 ± 319 | 1.4 ± 0.3 |
| 2017-11-01 | 305 | 0.27 ± 0.07 | 388 | 2.10 ± 0.11 | 1.48 ± 0.56 | 575 ± 217 | 0.6 ± 0.2 |
| 2017-11-16 | 320 | 0.17 ± 0.05 | 241 | 2.09 ± 0.10 | 0.91 ± 0.38 | 220 ± 92 | 0.2 ± 0.1 |
| 2017-11-18 | 322 | 0.54 ± 0.09 | 546 | 2.09 ± 0.10 | 2.90 ± 0.80 | 1585 ± 434 | 1.6 ± 0.4 |
| 2017-12-05 | 339 | 1.36 ± 0.15 | 373 | 2.09 ± 0.10 | 7.32 ± 1.56 | 2730 ± 583 | 2.7 ± 0.6 |
| 2017-12-06 | 340 | 0.33 ± 0.08 | 922 | 2.09 ± 0.10 | 1.79 ± 0.61 | 1655 ± 565 | 1.7 ± 0.6 |
| 2017-12-09 | 343 | 0.24 ± 0.08 | 407 | 2.09 ± 0.10 | 1.30 ± 0.57 | 530 ± 233 | 0.5 ± 0.2 |
| 2017-12-09 | 343 | 0.91 ± 0.13 | 390 | 2.09 ± 0.10 | 4.94 ± 1.20 | 1925 ± 469 | 1.9 ± 0.5 |
| 2018-01-22 | 22 | 0.84 ± 0.12 | 387 | 2.08 ± 0.10 | 4.53 ± 1.11 | 1752 ± 430 | 1.8 ± 0.4 |
| 2018-02-05 | 36 | 0.70 ± 0.11 | 146 | 2.07 ± 0.10 | 3.69 ± 0.93 | 538 ± 135 | 0.5 ± 0.1 |
| 2018-03-24 | 83 | 1.56 ± 0.16 | 250 | 2.11 ± 0.11 | 8.61 ± 1.75 | 2153 ± 437 | 2.2 ± 0.4 |
| 2018-04-12 | 102 | 0.70 ± 0.11 | 252 | 2.19 ± 0.11 | 4.16 ± 1.04 | 1048 ± 262 | 1.0 ± 0.3 |
| 2018-04-21 | 111 | 1.69 ± 0.17 | 292 | 2.23 ± 0.11 | 10.36 ± 2.08 | 3026 ± 608 | 3.0 ± 0.6 |
| 2018-04-29 | 119 | 0.96 ± 0.13 | 458 | 2.27 ± 0.11 | 6.09 ± 1.41 | 2787 ± 645 | 2.8 ± 0.6 |
| 2018-05-02 | 122 | 0.26 ± 0.08 | 531 | 2.28 ± 0.11 | 1.70 ± 0.67 | 905 ± 358 | 0.9 ± 0.4 |
| 2018-05-20 | 140 | 1.05 ± 0.13 | out of FOV | 2.37 ± 0.12 | 7.26 ± 1.64 | NA | NA |
| 2018-05-31 | 151 | 0.16 ± 0.06 | 309 | 2.42 ± 0.12 | 1.13 ± 0.55 | 349 ± 169 | 0.3 ± 0.2 |
| 2018-06-02 | 153 | 0.47 ± 0.09 | 413 | 2.42 ± 0.12 | 3.39 ± 1.00 | 1401 ± 414 | 1.4 ± 0.4 |
| 2018-06-04 | 155 | 0.22 ± 0.06 | 677 | 2.43 ± 0.12 | 1.63 ± 0.62 | 1102 ± 418 | 1.1 ± 0.4 |
| 2018-06-16 | 167 | 0.20 ± 0.06 | 291 | 2.46 ± 0.12 | 1.50 ± 0.58 | 437 ± 170 | 0.4 ± 0.2 |
| 2018-06-18 | 169 | 0.32 ± 0.15 | 382 | 2.47 ± 0.12 | 2.38 ± 1.11 | 908 ± 423 | 0.9 ± 0.4 |
| 2018-07-04 | 185 | 0.40 ± 0.08 | 340 | 2.48 ± 0.12 | 3.05 ± 0.94 | 1039 ± 320 | 1.0 ± 0.3 |
| 2018-07-25 | 206 | 0.33 ± 0.13 | out of FOV | 2.44 ± 0.12 | 2.42 ± 1.21 | NA | NA |
| 2018-07-28 | 209 | 1.94 ± 0.18 | 444 | 2.44 ± 0.12 | 14.21 ± 2.78 | 6309 ± 1233 | 6.3 ± 1.2 |
| 2018-09-21 | 264 | 1.90 ± 0.19 | out of FOV | 2.20 ± 0.11 | 11.34 ± 2.25 | NA | NA |





| 2018-09-21 | 264 | $0.13 \pm 0.11$ | out of FOV | $2.20 \pm 0.11$ | $0.78 \pm 0.69$ | NA | NA |
|---|---|---|---|---|---|---|---|
| 2018-10-02 | 275 | $0.23 \pm 0.08$ | out of FOV | $2.16 \pm 0.11$ | $1.31 \pm 0.62$ | NA | NA |
| 2018-10-04 | 277 | $0.59 \pm 0.10$ | 276 | $2.15 \pm 0.11$ | $3.38 \pm 0.90$ | $932 \pm 248$ | $0.9 \pm 0.2$ |
| 2018-10-04 | 277 | $0.86 \pm 0.12$ | 345 | $2.15 \pm 0.11$ | $4.95 \pm 1.17$ | $1709 \pm 405$ | $1.7 \pm 0.4$ |
| 2018-10-13 | 286 | $1.24 \pm 0.15$ | 914 | $2.13 \pm 0.11$ | $6.94 \pm 1.50$ | $6346 \pm 1375$ | $6.3 \pm 1.4$ |
| 2018-10-14 | 287 | $1.35 \pm 0.16$ | out of FOV | $2.13 \pm 0.11$ | $7.55 \pm 1.63$ | NA | NA |
| 2018-11-04 | 308 | $1.23 \pm 0.15$ | 915 | $2.09 \pm 0.10$ | $6.66 \pm 1.45$ | $6097 \pm 1331$ | $6.1 \pm 1.3$ |
| 2018-11-04 | 308 | $0.30 \pm 0.08$ | 609 | $2.09 \pm 0.10$ | $1.61 \pm 0.61$ | $983 \pm 372$ | $1.0 \pm 0.4$ |
| 2018-11-08 | 312 | $0.68 \pm 0.10$ | out of FOV | $2.09 \pm 0.10$ | $3.69 \pm 0.93$ | NA | NA |
| 2018-11-08 | 312 | $0.55 \pm 0.09$ | out of FOV | $2.09 \pm 0.10$ | $2.99 \pm 0.80$ | NA | NA |
| 2018-11-16 | 320 | $0.12 \pm 0.12$ | out of FOV | $2.09 \pm 0.10$ | $0.63 \pm 0.70$ | NA | NA |
| 2018-11-16 | 320 | $0.92 \pm 0.13$ | 292 | $2.09 \pm 0.10$ | $4.97 \pm 1.19$ | $1450 \pm 346$ | $1.4 \pm 0.3$ |
| 2019-01-11 | 11 | $0.57 \pm 0.11$ | 421 | $2.10 \pm 0.11$ | $3.08 \pm 0.93$ | $1297 \pm 392$ | $1.3 \pm 0.4$ |
| 2019-02-22 | 53 | $1.07 \pm 0.14$ | 318 | $2.07 \pm 0.10$ | $5.64 \pm 1.28$ | $1794 \pm 409$ | $1.8 \pm 0.4$ |
| 2019-02-24 | 55 | $1.61 \pm 0.16$ | out of FOV | $2.07 \pm 0.10$ | $8.51 \pm 1.70$ | NA | NA |
| 2019-03-02 | 61 | $1.69 \pm 0.17$ | 306 | $2.07 \pm 0.10$ | $8.95 \pm 1.77$ | $2739 \pm 543$ | $2.7 \pm 0.5$ |
| 2019-03-30 | 89 | $0.31 \pm 0.11$ | 453 | $2.13 \pm 0.11$ | $1.75 \pm 0.80$ | $793 \pm 364$ | $0.8 \pm 0.4$ |
| 2019-04-07 | 97 | $1.28 \pm 0.17$ | clouds | $2.17 \pm 0.11$ | $7.39 \pm 1.72$ | NA | NA |
| 2019-05-09 | 129 | $0.39 \pm 0.13$ | out of FOV | $2.32 \pm 0.12$ | $2.56 \pm 1.14$ | NA | NA |
