# Peer review of "Gravity wave instability structures and turbulence from more than one and a half years of OH\* airglow imager observations in Slovenia"

_Atmospheric Measurement Techniques, 2021_

## Referee Comment (RC1)

**Summary**

While this is an interesting data set in my opinion the authors have made significant errors in their analysis that makes it unacceptable in its present form. I do think though that the data set is potentially valuable, and I would urge the authors to rework their text, based on the comment I have provided, as a study of the characteristics of instabilities.

The major problems are as follows. 1) it is unlikely that any significant portion (or possibly none) of the features they are observing, at wavelengths below 5 km, are gravity waves (GWs). It is most likely they are instability features commonly seen in airglow images. 2) the presentation of the K and epsilon data is not clear in its present form. This has to be more convincing and warrants a larger discussion as the values are much higher than expected, or found in other studies. They also need to read the literature they cite.

**Are the seeing any gravity waves (GWs)?**

This study uses a high-spatial and temporal resolution imager that has 24 m pixels at the airglow layer. Unfortunately, to achieve the spatial resolution they restrict the FOV to just under 10 km. The data from this instrument have been analyzed, using 2D FFTs, to indicate the presence of wavelike features with horizontal wavelengths from 48 m to 4.5 km. They claim these are GWs although they do allow that some could be instabilities. They note that they could be advected by the wind as secondary gravity waves. The following are issues I have.

1. If they read the literature they cited, notably the Nakamura (1999) study and the series of Hecht papers (say the Rev of Geophysics review paper and the 2012 paper) it would immediately become obvious that current thinking is that features with scale sizes below 10 km are probably instability features caused either by the breakdown of an existing GW or instability features, such as KHIs, that are routinely formed in the airglow due to the formation of large wind shears. Now, while the authors do seem to imply that features with periods below the BV period are not GWs, the authors seem to argue that features that have periods above the BV period are GWs. But a 4 km wavelength instability feature moving with the wind at 5 m/s would have an observed period of 800 s, well above the BV period. So, all the features could easily be instabilities despite having long apparent periods.
2. Instability features are blown by the wind. GWs in general do not travel in the wind direction. But when they do two effects occur that make them less likely to be observed in the airglow layer. They are both related to the dispersion relation that shows that as the intrinsic velocity (the velocity with respect to the background wind) approaches the wave velocity the vertical wavelength decreases because the intrinsic frequency becomes small. This causes two effects.
    a. Unless the intrinsic frequency is very close to the BV frequency the vertical wavelength will be less than the horizontal wavelength. Now for the waves that are presented in this paper the vertical wavelength will be 4.5 km or much

smaller. GWs that have wavelengths thinner than the airglow layer (8-10 km) will suffer phase cancellation and will have vastly reduced amplitudes and likely will be difficult to see (Swenson and Liu,1998).

    b. As the vertical wavelength decreases the waves undergoes viscous dissipation and instability formation. This is discussed somewhat in Hecht et al., 2000 as well in Hecht et al., 2018. For the former and assuming the very large viscosity implied by the current work GW, lifetimes could seconds to a few minutes for the GWs in this study.

Related to b is that if the features are really blown by the wind they are at a critical level and probably do not survive. I should note that waves in this study are travelling at a very low speed so it is very likely that extremely common wind variations would exceed the wave speed and the critical level interaction (viscous dissipation or instability formation) would occur. Hence, it seems very unlikely these are GWs.

3. The characteristics of these waves if they are GWs, as currently presented, seem strange. Their phase speeds are quite low-below 20 m/s. GW climatology's typically show phase speeds of up to 50 m/s with the histogram of speeds centered closer to 40 m/s.

4. I was somewhat curious on how the monochromatic wavelengths were derived. They state they use a 2D FFT. Now FFTs assume the wave is present over the whole field of view and are often a little misleading with respect to monochromatic waves for airglow images because waves may be present over only a small fraction of the field. In the Hannawald reference they give a very nice image showing waves and I believe the FFT approach should be appropriate for date like that. But to date, while small scale instabilities have been identified with horizontal wavelength of a few to ~10 km there have been no reports of GWs with horizontal wavelengths of 5 km to 0.05 km. I would like to see images with their respective FFTs for images where the wavelengths are ~ 4 ,1,0.5,0.1 and 0.05 km. I am wondering if most of those images show features that resemble OH images (shown in the Hecht references) with instability features and their associated secondary instabilities and the resulting turbulences. I am really curious about GWs (or even instabilities/wave trains) with wavelengths at or much below ~500 m. These have not been reported before.

**Are the derived K and epsilon values realistic?**

This paper argues that they are seeing rotating cylinders of turbulence. Using a formulism developed by Prolss they proceed to analyze their data for the eddy diffusion, K (from Prolss) and energy dissipation rate, epsilon (using a Weinstock formula). There are two issues. The first is whether they are using the right formulae and measurables. The second is whether the rotating cylinder is the correct geometry.

**Deriving K and epsilon from airglow images**

In this paper they use the following formula to derive K as ~0.1Lv where L=2R, R is the radius of a rotating cylindrical tube and v is the rotation velocity. This is based on a Prolss 1961 analysis that I have not read. The energy dissipation rate epsilon is given by $KN^2$ where N is the BV frequency. Based on these relations they assert the following:

"The derived values of eddy diffusion coefficients are in the range around $10^3 – 10^4$ m²/s and agree mostly with earlier results from rocket and lidar measurements and simulations. Considering the respective values of the BV frequency as calculated by Wüst et al. (2020) we retrieve energy dissipation rates between 0.63 W/kg and 14.21 W/kg, that cause estimated heatings by 0.2 - 6.3 K per turbulence event. These have the same order of magnitude as the daily chemical heating rates as reported by Marsh (2011)."

There are a number of issues with this statement.

1. Recently there have been several attempts to derive the energy dissipation rate, epsilon, based on techniques and formulae that have been applied to radar images (Chau et al. 2020), TMA releases (Mesquita et al. 2020) and airglow imaging (Hecht et al, 2021). The Chau reference provides a formula for epsilon=$v^3$/L. Here v is the root mean square horizontal velocity and L is horizontal scale size. In the Chau paper they derive an epsilon of about 1 W/kg and they note that this is quite high compared to rocket measurements, essentially contradicting the statement in this paper (that has no references). Hecht et al. 2021 also derive an epsilon of ~ 1W/kg using airglow data.
2. With respect to #1 Hocking(1999) provides a good discussion (as does Chau) of a generalized approach to the dependence of K and epsilon on the measured parameters, v and L. There is no particular need to assume a rotating model as used in this paper. See eqns 14-15 in Hocking and also Weinstock for the relations between K and epsilon, and between N and L and v (N~6.8v/L), and see Chu for the constant (~1) in eqn 14 (epsilon= $v^3$/L). One important point is that N is not a constant that one can take from a climatology such as Wust. N varies due to the temperature gradient that can be quite steep in either direction. N thus could be significantly larger or smaller than climatology. It is 0 when the lapse rate is the adiabatic lapse rate. At other times, say for shear instabilities, often N is larger than the background. Fortunately, one doesn't need the T profile to calculate epsilon, only L and v both of which can be obtained from airglow images. That is the approach followed in Hecht et al., 2021. However, while L is relatively easy to see from the airglow images v is more difficult as the background wind velocity must be subtracted, and some estimate of a root mean square velocity must be made. Hecht et al., 2021 provide one approach to this problem and suggest some uncertainties.
3. The statement that the K and epsilon values derived in this work, which I think are not accurate, are consistent with the literature is misleading. The current values I believe are

too high. (see Hocking 1999 Figure 8 for another plot of measured epsilon where values well above 1 W/kg are not there). Also, several additional studies suggest background atmosphere K values below 100 $m^2$/s (see Hecht et al., 2018, 2021, and Guo et al., 2017).

**Are the turbulent motions best represented by 3D rotating cylinders?**

Looking at the video quickly it is easy to imagine a 3D rotating cylinder as indicated in Figure 3. However, looking more carefully many of the features seem to swirl around and grow and fade in brightness, in 2D, all which may lead the brain to interpret the motion as rotating in 3D. What complicates this interpretation is that the image plane is at an angle so we could be seeing motion predominantly in 2D as opposed to 3D. While the rotating cylinder model might apply to some features it is unlikely this is valid for most of the turbulence in their images.

I think the best approach would be to follow the approach of Chau and assume that the mean velocity in any direction scales as the feature size in that direction. Then they  would need to measure the scale size of the feature and the mean velocity associated with that feature. That will not be easy if they don't have a way of measuring the mean wind. If they have images with instability features, they can used to track the mean wind and then they could try to follow the approach of Hecht et al (2021) to retrieve v and L.  However, even that approach can lead to uncertainties especially in epsilon since that goes as $v^3$.

I also suspect that some of their images are just showing the result of (larger scale) gravity wave breakdown where just turbulence features, but no distinct wavelike features, are observed. In that case, they could try to estimate the mean wind by the mean motion of all the turbulence, and then calculate v from the  parcel velocity deviations from the mean wind for a particular eddy parcel.

Chau, J. L., Urco, J. M., Avsarkisov, V., Vierinen, J. P., Latteck, R., Hall, C. M., & Tsutsumi, M. (2020). Four-dimensional quantification of Kelvin-Helmholtz instabilities in the polar summer mesosphere using volumetric radar imaging. *Geophysical Research Letters*, **47**, e2019GL086081.

Guo, Y., Liu, A. Z., & Gardner, C. S. (2017).  First Na lidar measurements of turbulence heat flux, thermal diffusivity, and energy dissipation rate in the mesopause region. *Geophysical Research Letters*, **44**, 5782– 5790

Hecht, J. H.,  Fritts, D. C.,  Gelinas, L. J.,  Rudy, R. J.,  Walterscheid, R. L., &  Liu, A. Z. (2021).  Kelvin-Helmholtz billow interactions and instabilities in the mesosphere over the Andes Lidar Observatory: 1. Observations. *Journal of Geophysical Research: Atmospheres*,  126, e2020JD033414. https://doi.org/10.1029/2020JD033414

Hocking, W.K. The dynamical parameters of turbulence theory as they apply to middle atmosphere studies. *Earth Planet Sp* **51,** 525–541 (1999). https://doi.org/10.1186/BF03353213

Mesquita, R. L. A., Larsen, M. F., Azeem, I., Stevens, M. H., Williams, B. P., Collins, R. L., & Li, J.(2020).  In situ observations of neutral shear instability in the statically stable high-latitude mesosphere and lower thermosphere during quiet geomagnetic conditions. *Journal of Geophysical Research: Space Physics*,  **125**,  e2020JA027972. https://doi.org/10.1029/2020JA027972

Swenson, G. R., & Liu, A. Z. (1998).  A model for calculating acoustic gravity wave energy and momentum flux in the mesosphere from OH airglow. *Geophysical Research Letters*,  **25**(4),  477– 480. https://doi.org/10.1029/98GL00132

---

## Author Comment (AC1)

**Authors' Response to the Comment of Referee #1**

We thank Referee #1 for his detailed review and his suggestions for improvement.

Referee #1 identified two major problems.

1) It is unlikely that the small-scale wave structures we observe are gravity waves. They should rather be considered as instability features.

2) Our values of K and $\epsilon$ are too high and he doubts whether our approach of a rotating cylinder is the correct model for our turbulence observations.

**Passages in red have been deleted or rephrased as indicated.**

**Referring to major problem 1):**

We revised the discussion of our observed wave structures and switched to focus of our interpretation from gravity waves to instability features. The word 'wave' we use for convenience refers to wave-like features as seen in the images. We introduced section 4.1 with "We inserted the passage "Please note that we a using the word 'wave' for all wave-like structures we find in the images. The question whether these are actual gravity waves is discussed in section 5." at the beginning of section 4.1." to clarify this.

Referee #1 doubted that any significant portion of our observed structures with horizontal wavelengths below 5 km would be gravity waves. He gave multiple reasons why these structures would rather be instability features of gravity waves. We agree with his detailed explanations and refrain from considering wave structures with periods above the BV period gravity waves. As Referee #1 states correctly, Doppler-shifting may play a vital role and we have no systematic wind data to account for that.

We omitted the distinction between waves above and below the BV period in Figure 1. Consequently, also the sentence "The contribution of wave events with a period longer than the respective BV period is coloured in grey." disappeared in the caption of Figure 1. We updated the wording and the statistical values in section 4.1 (results: statistics of wave parameters) Lines 165 ff. resulting from this.

*"**Instability features** are blown by the wind. GWs in general do not travel in the wind direction. But when they do two effects occur that make them **less likely to be observed** in the airglow layer. They are both related to the **dispersion relation** that shows that as the intrinsic velocity (the velocity with respect to the background wind) approaches the wave velocity the vertical wavelength decreases because the intrinsic frequency becomes small. This causes two effects.*

> *Unless the intrinsic frequency is very close to the BV frequency the vertical wavelength will be less than the horizontal wavelength. Now for the waves that are presented in this paper the **vertical wavelength will be 4.5 km or much smaller**. GWs that have wavelengths thinner than the airglow layer (8-10 km) will suffer phase cancellation and will have vastly reduced amplitudes and likely will be difficult to see (Swenson and Liu,1998).*

> *As the vertical wavelength decreases the waves undergoes viscous dissipation and instability formation. This is discussed somewhat in Hecht et al., 2000 as well in Hecht et al., 2018. For*

*the former and assuming the very large viscosity implied by the current work GW, **lifetimes could be seconds to a few minutes** for the GWs in this study. Related to b is that if the features are really blown by the wind they are at a critical level and **probably do not survive**. I should note that waves in this study are travelling at a very low speed so it is very **likely that extremely common wind variations would exceed the wave speed and the critical level interaction (viscous dissipation or instability formation) would occur**. Hence, it seems very unlikely these are GWs."*

We agree with this detailed discussion of the dispersion relation.

We added 'If Figure 3b was the phase speed distribution of gravity waves, it is likely that a majority of them would encounter critical levels somewhere and would not be observable in the OH* layer.' to the discussion.

*"The characteristics of these waves if they are GWs, as currently presented, seem strange. Their phase speeds are quite low-below 20 m/s. GW climatology's typically show phase speeds of up to 50 m/s with the histogram of speeds centered closer to 40 m/s."*

We included this point in the discussion.

Line 360ff: "The quite slow phase speeds (mean value 13.3 m / s ) are one hint for this as typical gravity wave phase speeds accumulate around 40 m / s (see, e.g., Wachter et al., 2015 and Wüst et al., 2018)."

*"I was somewhat curious on how the monochromatic wavelengths were derived. They state they use a 2D FFT. Now FFTs assume the wave is present over the whole field of view and are often a little misleading with respect to monochromatic waves for airglow images because waves may be present over only a small fraction of the field. In the Hannawald reference they give a very nice image showing waves and I believe the FFT approach should be appropriate for date like that. But to date, while small scale instabilities have been identified with horizontal wavelength of a few to ~10 km there have been no reports of GWs with horizontal wavelengths of 5 km to 0.05 km. I would like to see images with their respective FFTs for images where the wavelengths are ~ 4 ,1,0.5,0.1 and 0.05 km. I am wondering if most of those images show features that resemble OH images (shown in the Hecht references) with instability features and their associated secondary instabilities and the resulting turbulences. I am really curious about GWs (or even instabilities/wave trains) with wavelengths at or much below ~500 m. These have not been reported before."*

We have not found waves with horizontal wavelengths down to 0.05 km, but this is the short-scale limit of our analysis, corresponding to twice the spatial resolution of 2*24 m = 48 m.

It is true that assuming stationarity the 2D-FFT rather finds wave structures that extend over the entire image. We have added a sample image of the smallest structure we found with a horizontal wavelength of ca. 1.9 km with its 2d spectrum (new Figure 1) and it does extend over at least half of the image.

In fact, we observed and reported a small "wave-like" instability feature with a horizontal wavelength of 550 m several years ago (see Sedlak et al., 2016). However, we doubt that such a small wave packet of limited spatial extension would appear in the 2D-FFT. A 2-dimensional wavelet analysis could account for such non-stationary features in future work.

**Referring to major problem 2):**

Considering the referee's extensive argumentation, we agree that assuming a rotating cylindrical model, as we did, exhibits several weaknesses. Demanding a perfect rotating cylinder is a very strong assumption that may apply to some of the turbulent vortices, but definitely not to all of them. As Referee #1 states correctly, there are several non-cylindrical vortices besides the rotation cylinder in our example in Figure 3 and there is no particular need for a rotating model. Furthermore, we agree that using climatological values of N for our quite short-time and localized episodes is relatively coarse.

We decided to follow his advice and adapted the method of Hecht et al. (2021) following Chau et al. (2020). We refrained from determining K from a rotating cylinder model. Instead, we read the feature size L and the residual velocity $v_{res}$ from the image series and directly calculated $\varepsilon$ by using the equation $\varepsilon = C \frac{v_{res}^3}{L}$ with $C \approx 1$ (Hecht et al., 2021). As Referee #1 assumed correctly, this was not always easy for all our examples. Staying with the episodes where the derivation of $\varepsilon$ was possible with this method, our data basis reduced from 45 to 25 episodes. This changed the results of our correlation analysis with gravity wave activity: We now hardly see any significant correlation with the activity of gravity waves between 6 and 480 min.

Although the values are now a bit lower (the rotating cylinder model certainly served as an upper boundary value since it assumed maximum mixing; Referee #2 stated this quite correctly) some of them still exceed the limit of 1 W/kg with the maximum being 9 W/kg. We provide a careful discussion of the deviations from previous studies.

As the derivation of $\varepsilon$ is quite difficult and due to the variety of turbulent episodes we furthermore decided to show more of our turbulent observations. Besides our former video supplement from 4 November 2018, we now present three more video sequences.

**Changes made in the manuscript:**

**Abstract**

Lines 14, 38, 49, 55, 72, 77, 108, 186, 264, 269, 272, 279, and 312: We put commas before and behind "e.g.".

Line 16f: Added "instability features from breaking secondary waves"

Line 17: Replaced "originating from breaking primary waves" by "that were created" and dropped "(westward)" and "(summer)".

Line 20f: We changed "Furthermore, observations of turbulent vortices allowed the estimation of eddy diffusion coefficients in the UMLT from image sequences in 45 cases. Values range around $10^3 - 10^4 \mathrm{~m^2 s^{-1}}$ and mostly agree with literature. Turbulently dissipated energy is derived taking into account values of the Brunt-Väisälä frequency based on TIMED-SABER (Thermosphere Ionosphere Mesosphere Energetics Dynamics, Sounding of the Atmosphere using Broadband Emission Radiometry) measurements as presented by Wüst et al. (2020). Energy dissipation rates range between $0.63 \mathrm{~W~kg^{-1}}$ and $14.21 \mathrm{~W~kg^{-1}}$ leading to an approximated maximum heating of

[revised manuscript text omitted]

Line 168 f: We dropped "The median value of gravity wave periods is found at 517 s (8.6 min).".

Line 169: 17.6 m/s changed to 139.8 m/s; 7.9 m/s changed to 13.3 m/s

Line 170: 3.3 m/s changed to 10.3 m/s. 50.7 % (49.3 %) changed to 52.7 % (47.3 %). Dropped the word "gravity".

Line 171: Dropped the word "gravity".

Line 172: 5.4 m/s (5.3 m/s) changed to 9.4 m/s (8.2 m/s). 3.0 m/s (3.1 m/s) changed to 8.9 m/s (7.4 m/s).

Line 173: 53.8 % changed to 56.0 %. 46.2 % changed to 44.0 %.

Line 174: 47.9 % changed to 49.5 %.

Line 175: 52.1 % changed to 50.5 %. 44.4 % (53.5 %) changed to 42.0 % (55.7 %).

Line 176: 4.7 m/s (5.3 m/s) changed to 7.5 m/s (9.4 m/s).

Line 177: 3.0 m/s (3.2 m/s) changed to 7.1 m/s (8.7 m/s).

Lines 192 ff: We added: "To give an impression of the turbulent dynamics we observe, we present four of our turbulence episodes as video supplement. On 16 November 2017, 02:16 UTC, the turbulent breakdown of parts of an extended wave field can be observed (video 1). On 6 December 2017, 00:26 UTC, several fronts seem to be building up and form rotating vortices (video 2). This can be observed even clearer on 14 October 2018, 17:08 UTC, where the residual movement of turbulent features can be well recognized above the general background movement (video 3). On 4 November 2018, 19:18 UTC, breaking wave fronts seem to form rotating structures of nearly cylindrical shape, while these are accompanied by other turbulently moving eddies (video 4).

We estimate the turbulent energy dissipation rate $\epsilon$ using equation (1). However, in contrast to Chau et al. (2020) who used radar measurements, we only have horizontal information from our airglow imager. Hecht et al. (2021) demonstrate an approach how to apply equation (1) to purely horizontal airglow imager data, which we adapt to our observations in the following. The characteristic length scale $L$ can be read from the images by measuring the size of the turbulent features. The velocity scale is given by the residual velocity $v_{res}$ of these features. In our observations, they are part of larger instability features, which we assume to be advected by the background wind. We determine $v_{res}$ by reading the actual velocity of the turbulent features and subtracting the background movement $v_{bg}$ in the resulting direction. This is exemplarily shown in Figure 4. The two patches highlighted therein are both moving to the upper right direction but are approaching each other. This helps distinguishing background and residual movement."

and dropped:

"The eddy parameters needed for the calculation of the eddy diffusion coefficient (Eq. 1) are determined manually from the image series of the 45 observations of turbulence. It has to be kept in mind that we are deriving properties of a three-dimensional movement from two-dimensional data.

We assume the vortices to rotate in a perfect circular shape. The lateral expansion creates the impression of a rotating cylinder. Coherently moving structures give indication of the horizontal velocity vector. Unless the rotational axis is aligned perpendicular to the image plane, the three-dimensional vortex rotation manifests as more than one coherent structure that is moving against or overtaking each other (see e.g. Sedlak et al., 2016; Figure 6 therein). During data inspection we

noticed that the orientation of the rotational axis can be aligned in any direction. It tends to be parallel to the image plane when it evolves directly from the crests of a breaking wave. However, we could also observe eddies rotating around an axis aligned almost perpendicular to the image plane. An example of a rotating vortex within a FAIM 3 snapshot on 4 November 2018 at 19:36:41 UTC is displayed in Figure 3a. The rotational axis and the direction of rotation are marked therein and on an actual cylinder (Figure 3b) for clarification. Since it is very difficult to identify a vortex structure in a single picture we have attached a video sequence of this episode (Video 1). The vortex radius and velocity are read from the images. Besides measuring the vortex rotation, it has also to be taken care of the overall image: if additional to the eddy movement all structures in the FOV are moving into a common direction, this background motion has to be subtracted. In the example shown above the vortex is advected toward the left corner. The distance between camera and observed vortex is much larger than the expansion of the vortex along the rotation axis so that falsifications arising from different perspectives of the vortices can be neglected. This principle is illustrated in Figure 4. As the vortices are three-dimensional the alignment of the rotational axis should not affect the value of the vortex parameters in the images: it does not matter if the axis is aligned perpendicular, parallel or in any other angle to the image plane, the vortex size will be accessible from the two-dimensional projection of the image assuming circular eddy movement. The same holds for the circumferential velocity since both the radius and the circulation time remain unchanged. However, perfectly circular eddy rotation does not necessarily occur in nature. Deviations from circularity can lead to both over- and underestimation of vortex sizes depending on the vortex orientation. Since isotropy is one of the characteristic properties of turbulent movements one may presume that from a statistical point of view both cases occur equally so that no systematic error is made."

Lines 232ff: We changed

"As stated in section 3 we found 45 episodes of turbulence that allowed the derivation of the vortex radius and circumferential velocity. The resulting eddy diffusion coefficients $K$ are shown in **Fehler! Verweisquelle konnte nicht gefunden werden.**. We assume a general read-out error of $\pm 3$ pixels, which corresponds to a distance of $\pm 72$ m. The circumferential velocity is determined by reading the distance a patch on the rotating cylinder surface covers within an episode of at least ten images, which corresponds to a time span of 28 s. Thus, the circumferential velocity is estimated with an error of $\pm 2.6$ m s$^{-1}$. The arising uncertainties of $K$ are calculated following the rules of error propagation.

The values of $K$ range from 0.12 to $1.94 \cdot 10^4$ m$^2$s$^{-1}$. The mean value is $0.76 \cdot 10^4$ m$^2$s$^{-1}$"

to

"As stated in section 3 we found 25 episodes of turbulence that allowed the derivation of $L$ and $v_{res}$. Using equation (1), the energy dissipation rate is then calculated by $\epsilon = \frac{v_{res}^3}{L}$. The resulting values are shown in **Fehler! Verweisquelle konnte nicht gefunden werden.**. We assume a general read-out error of $\pm 3$ pixels, which corresponds to a distance of $\pm 72$ m. Velocities are determined by reading the distance a feature covers within an episode of at least ten images, which corresponds to a time span of 28 s. Thus, velocities are estimated with an error of $\pm 2.6$ m s$^{-1}$. The arising uncertainties of $\epsilon$ are calculated following the rules of error propagation.

The values of $\epsilon$ range from 0.08 to 9.03 W kg$^{-1}$. The median value is 1.45 W kg"

Lines 240ff: We dropped

"(standard deviation of $0.53 \cdot 10^4$ m$^2$s$^{-1}$) and we retrieve a median of $0.59 \cdot 10^4$ m$^2$s$^{-1}$. It is difficult to exactly quantify the error of manual parameter determination from the images. However, in this work we rather focus on the order of magnitude of $K$. When calculating $K$ two distance values are read from the images (one for the vortex size and one for the determination of the circumferential speed). Considering Eq. (1), a mistake of factor 10 is made for $K$ if these distances are misread by a factor of at least $\sqrt{10}$. The shortest (and therefore most difficult to determine) diameter in our analysed examples was 768 m. For an error of one order of magnitude of $K$ this distance must be misread as either shorter than 243 m or longer than 2428 m, i.e. a distance of 32 pixels must be wrongly interpreted as shorter than 9 pixels or longer than 101 pixels. This lies far beyond the read-out uncertainty of $\pm 3$ pixels we introduced above and can be assumed to be much worse than any read-out error one would normally make.

The energy dissipation rate $\epsilon$ can be estimated from the eddy diffusion coefficient $K$ according to Eq. (2) using the BV frequency as described by Eq. (3).

As can be seen in **Fehler! Verweisquelle konnte nicht gefunden werden.** the energy dissipation rate of the observed turbulence events is in the range $0.63 - 14.21$ W kg$^{-1}$."

Line 254: We updated "146 s" to "241 s" and "2.4" to "4.0".

Line 258: We updated "220" to "30" and "6346" to "3015".

Line 262: We updated "0.2-6.3 K" to "0.03-3.02 K" and dropped "64 % (21 out of 33) of these values are larger than one Kelvin."

Line 268: We inserted a missing "°"

Lines 277ff: We changed "We find a slight but significant anticorrelation for gravity wave periods in the range $122 - 207$ min. For these periods the mean value of the correlation coefficient is -0.46. The highest coefficient of anticorrelation is -0.52 at a period of 178 min. "

to

"We find almost no significant correlation for any gravity wave period."

**Discussion:**

Line 280: We replaced "The directions of propagation are quite uniformly distributed over all quadrants as can be seen in Figure 2" by "As can be seen in Figure 3, the wave structures we observed exhibit multiple directions."

Line 288: We replaced "(positive zonal phase speed) and in westward direction during summer. Although this tendency is quite weak," by "whereas zonal directions are quite balanced during summer. Although the eastward tendency during winter is quite weak,".

Line 290: We added "tropospheric and".

Lines 292ff: We replaced "The reversed stratospheric winds during summer would consequently allow some more eastward travelling gravity waves to propagate upward (see e.g. Hoffmann et al., 2010; Hannawald et al., 2019). Since the highest observed phase speed of waves with periods longer than the BV period is only 17.6 m s-1, it can be assumed that in the majority we do not observe gravity waves that are originating from low altitudes and are fast enough not to be blocked by the stratospheric wind fields."

by

"During summer the stratospheric winds reverse to westward direction, so that eastward oriented gravity waves are filtered in the tropopause and westward oriented gravity waves are filtered in the stratosphere (see, e.g., Hoffmann et al., 2010; Hannawald et al., 2019).".

We inserted

"As we have no accompanying wind measurements in the height of our observations it is difficult to decide by means of the period whether the wave structures presented in section 4.1 are small-scale gravity waves or instability features. Ca. 63 % of the observed wave events have an observed period above the BV period (here we used the climatology presented by Wüst et al., 2020), however these could also be Doppler-shifted instability features instead of gravity waves. While the distinction between largely extended wave-fields (bands) and small localized wave structures that are related to instability (ripples) is often made at a horizontal wavelength of 10 - 20 km (Taylor et al., 1997; Nakamura et al., 1999), Li et al. (2017) remark that even structures with horizontal wavelengths of 5 - 10 km may sometimes be gravity waves rather than instability features."

after this passage.

Line 302 ff: We changed "Considering the directional distribution, it is possible that the major part of our waves may" to "If this would be true for our small-scale wave structures, they might rather".

Line 321 f: We changed "southward in 62 % of cases." to "southward in 71 % of cases.".

We inserted

"However, regarding the small horizontal wavelengths below 4.5 km, it is more likely that the major part of the observations presented in section 4.1 are related to instability features. The quite slow phase speeds (mean value 13.3 m / ) are one hint for this as typical gravity wave phase speeds accumulate around 40 m / s (see, e.g., Wachter et al., 2015 and Wüst et al., 2018). If Figure 3b was the phase speed distribution of gravity waves, it is likely that a majority of them would encounter critical levels somewhere and would not be observable in the OH* layer."

after this passage.

Line 371 ff: We replaced "Li et al. (2017) report that ripples are hard to distinguish from small-scale gravity waves. Height-resolved measurements of the horizontal wind would be needed to determine the local wind shear and make a profound statement about atmospheric instability. Nevertheless,"

by

"Considering the fact that the directional peculiarities of our observed wave events fit well with the expected behavior of secondary gravity waves, as discussed above, support the scenario of the wave structures being ripples from dynamic instabilities of secondary gravity waves, that originate from the stratospheric and mesospheric jet.".

We added

"Nevertheless, height-resolved measurements of the horizontal wind would be needed to determine the local wind shear and make a profound statement about atmospheric instability. It has to be kept in mind that a 2d-FFT was used. Thus, periodic structured are assumed to be stationary, i.e., they extend over the entire image. Faint structures that appear only in small parts of the image (as does for example the 550m wave packet in Sedlak et al., 2016; Fig. 2) would be underrepresented by this analysis."

at the end of the discussion section of the wave statistics.

Lines 384ff:

We changed

"There are still very few measurements of turbulent eddy diffusion coefficients in the UMLT. Lübken (1997) reports $K$ to be around $10^1 - 10^2$ m$^2$s$^{-1}$ at a height of 87 km at high latitudes. Hodges (1969) states that the eddy diffusion coefficient caused by gravity waves is typically around $10^3$ m$^2$s$^{-1}$. According to the CIRA (Committee on Space Research (COSPAR) International Reference Atmosphere) climatology of 1986 (NASA National Space Science Data Center, 2007) global values range between magnitudes of $10^2$ and $10^3$ m$^2$s$^{-1}$. LIDAR measurements above New Mexico, USA deliver values that vary strongly around a magnitude of $10^2$ m$^2$s$^{-1}$ (Liu, 2009). Smith (2012) notes that the WACCM (Whole Atmosphere Community Climate Model) climatology exhibits rather small values with magnitude $10^1$ m$^2$s$^{-1}$ and that the huge discrepancies of $K$ estimates cannot be fully explained yet. The here-presented values of $K$ exhibit a magnitude of $10^3 - 10^4$ m$^2$s$^{-1}$, which partly agrees with recent results, although some of our values are higher."

to

"Measuring the eddy diffusion coefficient in the UMLT is still challenging and there are only few studies yet. Rocket measurements of Lübken (1997) deliver energy dissipation rates between ca. 0.01 and 0.1 W kg$^{-1}$ between 85 and 90 km height at high latitudes. Chau et al. (2020) find an energy dissipation rate of 1.125 W kg$^{-1}$ for their KHI event observed in the summer mesopause and state that this a rather high value compared to the findings of Lübken et al. (2002). Hocking (1999) provides a rescaled overview of earlier values of the energy dissipation rate and these have a maximum magnitude of 0.1 W kg$^{-1}$. Hecht et al. (2021) derive a value of 0.97 W kg$^{-1}$ from airglow images of a KHI event. Ranging from 0.08 up to 9.03 W kg$^{-1}$ the values of energy dissipation rate derived here are higher than reported by other studies. However, the median value of 1.45 W kg$^{-1}$ is not too far away from the values of Chau et al. (2020) and Hecht et al. (2021).".

Lines 402f: We added "– except for the studies of Hecht et al. (2021), whose value is quite similar to the median value of our data –„

Lines 405ff: We dropped "Nevertheless, the agreement of the above-mentioned authors on eddy parameters in the UMLT is quite good, considering the fact that energy dissipation rate in the upper troposphere and lower stratosphere varies by a factor of more than five orders of magnitude (Li et al., 2016).".

Line 408: Changed "vortex" to "turbulence"

Line 409: Changed "Circumferential speed and vortex radius" to "The length scale and velocity scale of turbulent features"

Line 410: "and we" changed to ". We"

Lines 412f: Inserted "However, using equation (1) velocity dominates the length scale due to its power of 3, so that $\epsilon$ strongly depends on a parameter, which is quite difficult to extract from the images."

Line 414: Dropped "eddy diffusion coefficient and"

Line 415: Replaced "eddy diffusion coefficients" by "energy dissipation rate" and replaced "significant anticorrelation" by "no significant correlation"

Lines 416: Changed "122-207 min" to "6-480 min"

Lines 416ff: Changed

[revised manuscript text omitted]

**Appendix**

We dropped Appendix A.

**Figures**

Figure 1 is now a sample event with its 2D-FFT spectrum.

Figure 2 (former Figure 1): Histograms of wave parameters without separation by the BV period

Figure 3 is now former Figure 2

Figure 4 (former Figure 3): New snapshot showing how to read the new wave parameters from the images.

Former Figure 4: dropped

Figure 5: Plot of temporal course and histogram of $\epsilon$ (no K values anymore)

Figure 6: Updated histogram of temperature changes

Figure 7: Updated plot of correlation analysis

Figure 8 (concept of rotating cylinder): dropped

**Tables**

We updated Table 1 by dropping those lines referring to events which we could not analyse anymore with the new method. We dropped the columns "DoY", "K" and "Angular BV Frequency".

---

## Author Response (AR1)

**Authors' Response to the Comment of Referee #1**

We thank Referee #1 for his detailed review and his suggestions for improvement.

Referee #1 identified two major problems.

1) It is unlikely that the small-scale wave structures we observe are gravity waves. They should rather be considered as instability features.

2) Our values of K and  $\epsilon$  are too high and he doubts whether our approach of a rotating cylinder is the correct model for our turbulence observations.

**Passages in red have been deleted or rephrased as indicated.**

**Referring to major problem 1):**

We revised the discussion of our observed wave structures and switched to focus of our interpretation from gravity waves to instability features. The word 'wave' we use for convenience refers to wave-like features as seen in the images. We introduced section 4.1 with "We inserted the passage "Please note that we a using the word 'wave' for all wave-like structures we find in the images. The question whether these are actual gravity waves is discussed in section 5." at the beginning of section 4.1." to clarify this.

Referee #1 doubted that any significant portion of our observed structures with horizontal wavelengths below 5 km would be gravity waves. He gave multiple reasons why these structures would rather be instability features of gravity waves. We agree with his detailed explanations and refrain from considering wave structures with periods above the BV period gravity waves. As Referee #1 states correctly, Doppler-shifting may play a vital role and we have no systematic wind data to account for that.

We omitted the distinction between waves above and below the BV period in Figure 1. Consequently, also the sentence "The contribution of wave events with a period longer than the respective BV period is coloured in grey." disappeared in the caption of Figure 1. We updated the wording and the statistical values in section 4.1 (results: statistics of wave parameters) Lines 165 ff. resulting from this.

**"Instability features** are blown by the wind. GWs in general do not travel in the wind direction. But when they do two effects occur that make them **less likely to be observed** in the airglow layer. They are both related to the **dispersion relation** that shows that as the intrinsic velocity (the velocity with respect to the background wind) approaches the wave velocity the vertical wavelength decreases because the intrinsic frequency becomes small. This causes two effects.

Unless the intrinsic frequency is very close to the BV frequency the vertical wavelength will be less than the horizontal wavelength. Now for the waves that are presented in this paper the **vertical wavelength will be 4.5 km or much smaller**. GWs that have wavelengths thinner than the airglow layer (8-10 km) will suffer phase cancellation and will have vastly reduced amplitudes and likely will be difficult to see (Swenson and Liu, 1998).

As the vertical wavelength decreases the waves undergoes viscous dissipation and instability formation. This is discussed somewhat in Hecht et al., 2000 as well in Hecht et al., 2018. For

the former and assuming the very large viscosity implied by the current work GW, **lifetimes** could be seconds to a few minutes for the GWs in this study. Related to b is that if the features are really blown by the wind they are at a critical level and probably do not survive. I should note that waves in this study are travelling at a very low speed so it is very likely that extremely common wind variations would exceed the wave speed and the critical level interaction (viscous dissipation or instability formation) would occur. Hence, it seems very unlikely these are GWs."

We agree with this detailed discussion of the dispersion relation.

We added 'If Figure 3b was the phase speed distribution of gravity waves, it is likely that a majority of them would encounter critical levels somewhere and would not be observable in the OH\* layer.' to the discussion.

"The characteristics of these waves if they are GWs, as currently presented, seem strange. Their phase speeds are quite low-below 20 m/s. GW climatology's typically show phase speeds of up to 50 m/s with the histogram of speeds centered closer to 40 m/s."

We included this point in the discussion.

Line 360ff: "The quite slow phase speeds (mean value 13.3 m / s ) are one hint for this as typical gravity wave phase speeds accumulate around 40 m / s (see, e.g., Wachter et al., 2015 and Wüst et al., 2018)."

"I was somewhat curious on how the monochromatic wavelengths were derived. They state they use a 2D FFT. Now FFTs assume the wave is present over the whole field of view and are often a little misleading with respect to monochromatic waves for airglow images because waves may be present over only a small fraction of the field. In the Hannawald reference they give a very nice image showing waves and I believe the FFT approach should be appropriate for date like that. But to date, while small scale instabilities have been identified with horizontal wavelength of a few to ~10 km there have been no reports of GWs with horizontal wavelengths of 5 km to 0.05 km. I would like to see images with their respective FFTs for images where the wavelengths are ~ 4 ,1,0.5,0.1 and 0.05 km. I am wondering if most of those images show features that resemble OH images (shown in the Hecht references) with instability features and their associated secondary instabilities and the resulting turbulences. I am really curious about GWs (or even instabilities/wave trains) with wavelengths at or much below ~500 m. These have not been reported before."

We have not found waves with horizontal wavelengths down to 0.05 km, but this is the short-scale limit of our analysis, corresponding to twice the spatial resolution of 2\*24 m = 48 m.

It is true that assuming stationarity the 2D-FFT rather finds wave structures that extend over the entire image. We have added a sample image of the smallest structure we found with a horizontal wavelength of ca. 1.9 km with its 2d spectrum (new Figure 1) and it does extend over at least half of the image.

In fact, we observed and reported a small "wave-like" instability feature with a horizontal wavelength of 550 m several years ago (see Sedlak et al., 2016). However, we doubt that such a small wave packet of limited spatial extension would appear in the 2D-FFT. A 2-dimensional wavelet analysis could account for such non-stationary features in future work.

**Referring to major problem 2):**

Considering the referee's extensive argumentation, we agree that assuming a rotating cylindrical model, as we did, exhibits several weaknesses. Demanding a perfect rotating cylinder is a very strong assumption that may apply to some of the turbulent vortices, but definitely not to all of them. As Referee #1 states correctly, there are several non-cylindrical vortices besides the rotation cylinder in our example in Figure 3 and there is no particular need for a rotating model. Furthermore, we agree that using climatological values of N for our quite short-time and localized episodes is relatively coarse.

We decided to follow his advice and adapted the method of Hecht et al. (2021) following Chau et al. (2020). We refrained from determining K from a rotating cylinder model. Instead, we read the feature size L and the residual velocity  $v_{res}$  from the image series and directly calculated  $\varepsilon$  by using the equation  $\varepsilon = C \frac{v_{res}^3}{L}$  with  $C \approx 1$  (Hecht et al., 2021). As Referee #1 assumed correctly, this was not always easy for all our examples. Staying with the episodes where the derivation of  $\varepsilon$  was possible with this method, our data basis reduced from 45 to 25 episodes. This changed the results of our correlation analysis with gravity wave activity: We now hardly see any significant correlation with the activity of gravity waves between 6 and 480 min.

Although the values are now a bit lower (the rotating cylinder model certainly served as an upper boundary value since it assumed maximum mixing; Referee #2 stated this quite correctly) some of them still exceed the limit of 1 W/kg with the maximum being 9 W/kg. We provide a careful discussion of the deviations from previous studies.

As the derivation of  $\varepsilon$  is quite difficult and due to the variety of turbulent episodes we furthermore decided to show more of our turbulent observations. Besides our former video supplement from 4 November 2018, we now present three more video sequences.

**Changes made in the manuscript:**

**Abstract**

Lines 14, 38, 49, 55, 72, 77, 108, 186, 264, 269, 272, 279, and 312: We put commas before and behind "e.g.".

Line 16f: Added "instability features from breaking secondary waves"

Line 17: Replaced "originating from breaking primary waves" by "that were created" and dropped "(westward)" and "(summer)".

Line 20f: We changed "Furthermore, observations of turbulent vortices allowed the estimation of eddy diffusion coefficients in the UMLT from image sequences in 45 cases. Values range around  $10^3 - 10^4 \text{ m}^2 \text{s}^{-1}$  and mostly agree with literature. Turbulently dissipated energy is derived taking into account values of the Brunt-Väisälä frequency based on TIMED-SABER (Thermosphere lonosphere Mesosphere Energetics Dynamics, Sounding of the Atmosphere using Broadband Emission Radiometry) measurements as presented by Wüst et al. (2020). Energy dissipation rates range between  $0.63 \text{ W kg}^{-1}$  and  $14.21 \text{ W kg}^{-1}$  leading to an approximated maximum heating of

0.2-6.3 K per turbulence event. These are in the same range as the daily chemical heating rates reported by Marsh (2011), which apparently stresses the importance of dynamical energy conversion in the UMLT."

**to**

"We present multiple observations of turbulence episodes captured by our high-resolution airglow imager and estimated the energy dissipation rate in the UMLT from image sequences in 25 cases. Values range around 0.08 and 9.03 W kg-1 and are higher than those in recent literature. The values found here would lead to an approximated localized maximum heating of 0.03-3.02 K per turbulence event. These are in the same range as the daily chemical heating rates for the entire atmosphere reported by Marsh (2011), which apparently stresses the importance of dynamical energy conversion in the UMLT."

**Introduction**

Line 40: We changed "ground" to "troposphere"

Lines 54ff: We changed

"They cause turbulent mixing of the medium, which is described by the eddy diffusion coefficient K. K can be calculated from the eddy radius  $r_e$  and the circumferential velocity  $v_e$  by

$$K = \frac{2}{3\pi} r_e v_e \tag{1}$$

(see e.g. Prölss, 2001. The derivation is outlined in detail in appendix A.). According to Weinstock (1978), knowing K and the BV frequency N, an estimate for the energy dissipation rate  $\epsilon$  - the rate at which turbulent kinetic energy is dissipated into heat at the short-scale end of the energy cascade of the inertial subrange (Li et al., 2016) - can be calculated using

$$K \approx 0.81 \cdot \left(\frac{\epsilon}{N^2}\right).$$
 (2)"

to

"They cause turbulent mixing of the medium, resulting in the dissipation of turbulent energy at an energy dissipation rate  $\epsilon$ . According to the theory of stratified turbulence,  $\epsilon$  depends on the characteristic length scale L and velocity scale U of the turbulent features. The energy dissipation rate is then given by

$$\epsilon = C_{\epsilon} \frac{U^3}{L} \tag{1}$$

(see, e.g., Chau et al, 2020; who apply this equation to radar observations of KHIs).  $C_{\epsilon}$  is a constant which is found to be equal to 1 (Gargett, 1999)."

**Lines 70ff: We changed**

"This is why turbulence investigations in the UMLT are challenging and there are only few values of K available at UMLT heights. Lübken et al. (1997) use rocket measurements to retrieve K and  $\epsilon$  in the height range 65-120 km. Liu (2009) presents a method for the estimation of K from gravity wave momentum fluxes derived from lidar data. Baumgarten & Fritts (2014) use imaging techniques of mesospheric noctilucent clouds to investigate the formation of KHIs and the onset of turbulence."

"This is why turbulence investigations in the UMLT are challenging and there are only few values of  $\epsilon$  available at UMLT heights. Lübken et al. (1997) use rocket measurements to retrieve  $\epsilon$  in the height range 65-120 km. Baumgarten & Fritts (2014) use imaging techniques of mesospheric noctilucent clouds to investigate the formation of KHIs and the onset of turbulence.".

Line 76: We changed "These include..." to "Remote sensing techniques include..."

Line 83: We changed "Proceedings" to "Improvements"

**Data Basis**

Line 138: We inserted "...from the input images..."during the 2d-FFT".

Line 156: We inserted "An exemplary event and the respective 2-dimensional spectrum are shown in Figure 1. We often observe episodes of turbulence in our image series that exhibit the typical dynamics of vortex formation and quasi-chaotic behavior."

Line 164: Replaced "45" by "25" and "vortex" by "turbulence"

Lines 166ff: Omitted "For both, gravity wave statistics (section 4.1) and the calculation of the energy dissipation rate (section 4.2) the BV frequency is required, which is adapted from the climatology presented by Wüst et al. (2020). It is based on TIMED-SABER (Thermosphere Ionosphere Mesosphere Energetics Dynamics, Sounding of the Atmosphere using Broadband Emission Radiometry) temperature data and takes into account the seasonal variability of the angular BV frequency. The climatology of the grid point (45° N, 10° E) is used, which is closest to our FOV. Depending on the day of the year (DoY) the BV frequency is given by

$$N = 2.20 \cdot 10^{-2} \mathrm{s}^{-2} + 0.19 \cdot 10^{-2} \mathrm{s}^{-2} \sin\left(\frac{2\pi}{324.51\mathrm{d}} \cdot DoY - 2.02\right) + 0.05 \cdot 10^{-2} \mathrm{s}^{-2} \sin\left(\frac{2\pi}{180.00\mathrm{d}} \cdot DoY + 1.61\right).$$
 (3)

We use an uncertainty of ±5% as according to Wüst et al. (2020) 91% of their data lie within this range around the harmonic approximation. The BV period is then referred to as  $\tau_{BV} = \frac{2\pi}{N}$ ."

**Results**

We inserted the passage "Please note that we a using the word 'wave' for all wave-like structures we find in the images. The question whether these are actual gravity waves is discussed in section 5." at the beginning of section 4.1.

Line 165 ff: We dropped "For each wave event the individual BV period is calculated based on Eq. (3) and the DoY. Ca. 63% of the observed wave events have a period longer than the respective BV period and will be referred to as gravity wave events in the following. Their statistical contribution is highlighted in grey in Fehler! Verweisquelle konnte nicht gefunden werden.."

Line 168 f: We dropped "The median value of gravity wave periods is found at 517 s (8.6 min).".

Line 169: 17.6 m/s changed to 139.8 m/s; 7.9 m/s changed to 13.3 m/s

to

Line 170: 3.3 m/s changed to 10.3 m/s. 50.7 % (49.3 %) changed to 52.7 % (47.3 %). Dropped the word "gravity".

Line 171: Dropped the word "gravity".

Line 172: 5.4 m/s (5.3 m/s) changed to 9.4 m/s (8.2 m/s). 3.0 m/s (3.1 m/s) changed to 8.9 m/s (7.4 m/s).

Line 173: 53.8 % changed to 56.0 %. 46.2 % changed to 44.0 %.

Line 174: 47.9 % changed to 49.5 %.

Line 175: 52.1 % changed to 50.5 %. 44.4 % (53.5 %) changed to 42.0 % (55.7 %).

Line 176: 4.7 m/s (5.3 m/s) changed to 7.5 m/s (9.4 m/s).

Line 177: 3.0 m/s (3.2 m/s) changed to 7.1 m/s (8.7 m/s).

[revised manuscript text omitted]

**Lines 240ff: We dropped**

"(standard deviation of  $0.53 \cdot 10^4 \text{ m}^2 \text{s}^{-1}$ ) and we retrieve a median of  $0.59 \cdot 10^4 \text{ m}^2 \text{s}^{-1}$ . It is difficult to exactly quantify the error of manual parameter determination from the images. However, in this work we rather focus on the order of magnitude of K. When calculating K two distance values are read from the images (one for the vortex size and one for the determination of the circumferential speed). Considering Eq. (1), a mistake of factor 10 is made for K if these distances are misread by a factor of at least  $\sqrt{10}$ . The shortest (and therefore most difficult to determine) diameter in our analysed examples was 768 m. For an error of one order of magnitude of K this distance must be misread as either shorter than 243 m or longer than 2428 m, i.e. a distance of 32 pixels must be wrongly interpreted as shorter than 9 pixels or longer than 101 pixels. This lies far beyond the readout uncertainty of  $\pm 3$  pixels we introduced above and can be assumed to be much worse than any read-out error one would normally make.

The energy dissipation rate  $\epsilon$  can be estimated from the eddy diffusion coefficient K according to Eq. (2) using the BV frequency as described by Eq. (3).

As can be seen in **Fehler! Verweisquelle konnte nicht gefunden werden.** the energy dissipation rate of the observed turbulence events is in the range  $0.63 - 14.21 W kg^{-1}$ ."

Line 254: We updated "146 s" to "241 s" and "2.4" to "4.0".

Line 258: We updated "220" to "30" and "6346" to "3015".

Line 262: We updated "0.2-6.3 K" to "0.03-3.02 K" and dropped "64% (21 out of 33) of these values are larger than one Kelvin."

Line 268: We inserted a missing "°"

Lines 277ff: We changed "We find a slight but significant anticorrelation for gravity wave periods in the range 122-207 min. For these periods the mean value of the correlation coefficient is -0.46. The highest coefficient of anticorrelation is -0.52 at a period of 178 min. "

**to**

"We find almost no significant correlation for any gravity wave period."

**Discussion:**

Line 280: We replaced "The directions of propagation are quite uniformly distributed over all quadrants as can be seen in Figure 2" by "As can be seen in Figure 3, the wave structures we observed exhibit multiple directions."

Line 288: We replaced "(positive zonal phase speed) and in westward direction during summer. Although this tendency is quite weak," by "whereas zonal directions are quite balanced during summer. Although the eastward tendency during winter is quite weak,".

Line 290: We added "tropospheric and".

Lines 292ff: We replaced "The reversed stratospheric winds during summer would consequently allow some more eastward travelling gravity waves to propagate upward (see e.g. Hoffmann et al., 2010; Hannawald et al., 2019). Since the highest observed phase speed of waves with periods longer than the BV period is only 17.6 m s-1, it can be assumed that in the majority we do not observe gravity waves that are originating from low altitudes and are fast enough not to be blocked by the stratospheric wind fields."

**by**

"During summer the stratospheric winds reverse to westward direction, so that eastward oriented gravity waves are filtered in the tropopause and westward oriented gravity waves are filtered in the stratosphere (see, e.g., Hoffmann et al., 2010; Hannawald et al., 2019).".

**We inserted**

"As we have no accompanying wind measurements in the height of our observations it is difficult to decide by means of the period whether the wave structures presented in section 4.1 are small-scale gravity waves or instability features. Ca. 63 % of the observed wave events have an observed period above the BV period (here we used the climatology presented by Wüst et al., 2020), however these could also be Doppler-shifted instability features instead of gravity waves. While the distinction between largely extended wave-fields (bands) and small localized wave structures that are related to instability (ripples) is often made at a horizontal wavelength of 10 - 20 km (Taylor et al., 1997; Nakamura et al., 1999), Li et al. (2017) remark that even structures with horizontal wavelengths of 5 - 10 km may sometimes be gravity waves rather than instability features."

**after this passage.**

Line 302 ff: We changed "Considering the directional distribution, it is possible that the major part of our waves may" to "If this would be true for our small-scale wave structures, they might rather".

Line 321 f: We changed "southward in 62 % of cases." to "southward in 71 % of cases.".

**We inserted**

"However, regarding the small horizontal wavelengths below 4.5 km, it is more likely that the major part of the observations presented in section 4.1 are related to instability features. The quite slow phase speeds (mean value 13.3 m / ) are one hint for this as typical gravity wave phase speeds accumulate around 40 m / s (see, e.g., Wachter et al., 2015 and Wüst et al., 2018). If Figure 3b was the phase speed distribution of gravity waves, it is likely that a majority of them would encounter critical levels somewhere and would not be observable in the OH\* layer."

**after this passage.**

Line 371 ff: We replaced "Li et al. (2017) report that ripples are hard to distinguish from small-scale gravity waves. Height-resolved measurements of the horizontal wind would be needed to determine the local wind shear and make a profound statement about atmospheric instability. Nevertheless,"

"Considering the fact that the directional peculiarities of our observed wave events fit well with the expected behavior of secondary gravity waves, as discussed above, support the scenario of the wave structures being ripples from dynamic instabilities of secondary gravity waves, that originate from the stratospheric and mesospheric jet.".

**We added**

"Nevertheless, height-resolved measurements of the horizontal wind would be needed to determine the local wind shear and make a profound statement about atmospheric instability. It has to be kept in mind that a 2d-FFT was used. Thus, periodic structured are assumed to be stationary, i.e., they extend over the entire image. Faint structures that appear only in small parts of the image (as does for example the 550m wave packet in Sedlak et al., 2016; Fig. 2) would be underrepresented by this analysis."

at the end of the discussion section of the wave statistics.

Lines 384ff:

**We changed**

"There are still very few measurements of turbulent eddy diffusion coefficients in the UMLT. Lübken (1997) reports K to be around  $10^1 - 10^2 \text{ m}^2 \text{s}^{-1}$  at a height of 87 km at high latitudes. Hodges (1969) states that the eddy diffusion coefficient caused by gravity waves is typically around  $10^3 \text{ m}^2 \text{s}^{-1}$ . According to the CIRA (Committee on Space Research (COSPAR) International Reference Atmosphere) climatology of 1986 (NASA National Space Science Data Center, 2007) global values range between magnitudes of  $10^2$  and  $10^3 \text{ m}^2 \text{s}^{-1}$ . LIDAR measurements above New Mexico, USA deliver values that vary strongly around a magnitude of  $10^2 \text{ m}^2 \text{s}^{-1}$  (Liu, 2009). Smith (2012) notes that the WACCM (Whole Atmosphere Community Climate Model) climatology exhibits rather small values with magnitude  $10^1 \text{ m}^2 \text{s}^{-1}$  and that the huge discrepancies of K estimates cannot be fully explained yet. The here-presented values of K exhibit a magnitude of  $10^3 - 10^4 \text{ m}^2 \text{s}^{-1}$ , which partly agrees with recent results, although some of our values are higher."

**to**

"Measuring the eddy diffusion coefficient in the UMLT is still challenging and there are only few studies yet. Rocket measurements of Lübken (1997) deliver energy dissipation rates between ca. 0.01 and 0.1W kg-1 between 85 and 90 km height at high latitudes. Chau et al. (2020) find an energy dissipation rate of 1.125 W kg-1 for their KHI event observed in the summer mesopause and state that this a rather high value compared to the findings of Lübken et al. (2002). Hocking (1999) provides a rescaled overview of earlier values of the energy dissipation rate and these have a maximum magnitude of 0.1 W kg-1. Hecht et al. (2021) derive a value of 0.97 W kg-1 from airglow images of a KHI event. Ranging from 0.08 up to 9.03 W kg-1 the values of energy dissipation rate derived here are higher than reported by other studies. However, the median value of 1.45 W kg-1 is not too far away from the values of Chau et al. (2020) and Hecht et al. (2021)."

Lines 402f: We added "– except for the studies of Hecht et al. (2021), whose value is quite similar to the median value of our data -,"

Lines 405ff: We dropped "Nevertheless, the agreement of the above-mentioned authors on eddy parameters in the UMLT is quite good, considering the fact that energy dissipation rate in the upper troposphere and lower stratosphere varies by a factor of more than five orders of magnitude (Li et al., 2016).".

Line 408: Changed "vortex" to "turbulence"

Line 409: Changed "Circumferential speed and vortex radius" to "The length scale and velocity scale of turbulent features"

Line 410: "and we" changed to ". We"

Lines 412f: Inserted "However, using equation (1) velocity dominates the length scale due to its power of 3, so that  $\epsilon$  strongly depends on a parameter, which is quite difficult to extract from the images."

Line 414: Dropped "eddy diffusion coefficient and"

Line 415: Replaced "eddy diffusion coefficients" by "energy dissipation rate" and replaced "significant anticorrelation" by "no significant correlation"

Lines 416: Changed "122-207 min" to "6-480 min"

**Lines 416ff: Changed**

[revised manuscript text omitted]

**Appendix**

We dropped Appendix A.

**Figures**

Figure 1 is now a sample event with its 2D-FFT spectrum.

Figure 2 (former Figure 1): Histograms of wave parameters without separation by the BV period

Figure 3 is now former Figure 2

Figure 4 (former Figure 3): New snapshot showing how to read the new wave parameters from the images.

Former Figure 4: dropped

Figure 5: Plot of temporal course and histogram of  $\epsilon$  (no K values anymore)

Figure 6: Updated histogram of temperature changes

Figure 7: Updated plot of correlation analysis

Figure 8 (concept of rotating cylinder): dropped

**Tables**

We updated Table 1 by dropping those lines referring to events which we could not analyse anymore with the new method. We dropped the columns "DoY", "K" and "Angular BV Frequency".

We thank Referee #2 for his valuable comments.

**Passages in red have been deleted or rephrased as indicated.**

His general comments were:

1. "I have some concerns about the concept of Prölss (see also the specific comments at the end). The idea is that the mixing occurs at time scales of half a rotation of the cylinder. This time directly affects the derived eddy diffusion coefficient. However, the rotation will not be over after half a rotation but will continue and the vortex will disintegrate into smaller vortices etc. It may well by that the assumption that the mixing occurs at time scales of half a rotation leads to a systematic underestimation of the effective mixing time and hence to an overestimation of the eddy diffusion coefficients. This may also explain why your values of K are systematically larger than the literature values cited."

We agree about the concerns Referee #2 has about the cylindrical concept of Prölss. Demanding a perfect rotating cylinder is a very strong assumption that may apply to some of the turbulent vortices but definitely not to all of them. We decided to follow his advice and adapted the method of Hecht et al. (2021) following Chau et al. (2020). We refrained from determining K from a rotating cylinder model. Instead, we read the feature size L and the residual velocity  $v_{res}$  from the image series and directly calculated  $\varepsilon$  by using the equation  $\varepsilon = C \frac{v_{res}^3}{L}$  with  $C \approx 1$  (Hecht et al., 2021). As Referee #1 assumed correctly, this was not always easy for all our examples. Staying with the episodes where the derivation of  $\varepsilon$  was possible with this method, our data basis reduced from 45 to 25 episodes. This changed the results of our correlation analysis with gravity wave activity: We now hardly see any significant correlation with the activity of gravity waves between 6 and 480 min.

Although the values are now a bit lower (the rotating cylinder model certainly served as an upper boundary value since it assumed maximum mixing; Referee #2 stated this quite correctly) some of them still exceed the limit of 1 W/kg with the maximum being 9 W/kg. We provide a careful discussion of the deviations from previous studies.

As the derivation of  $\varepsilon$  is quite difficult and due to the variety of turbulent episodes we furthermore decided to show more of our turbulent observations. Besides our former video supplement from 4 November 2018, we now present three more video sequences.

**2. "Please show a sample 2-D FFT spectrum."**

As Referee #2 desired, we now show a sample 2-D FFT in Figure 1. We also present the respective image, where the wave structure can be recognized.

Referring to his specific comments:

Line 37: We changed "ground" to "troposphere".

Line 68: We changed "These" to "Remote sensing techniques".

Line 74: We changed "Proceedings" to "Improvements".

Line 104: "automatic measurements with focus on the OH\* airglow"

"What does this mean (i.e. "with focus on")? Has a filter been used to remove the O2 singlet delta emission?"

No, we have not used any filter. Our formulation "with focus on" refers to the OH\* emissions dominating the spectral range between 0.9 and 1.7 µm where our camera system integrates. The influence of O2 singlet delta has been studied by Hannawald et al (2016), who found no significant influence of O2 singlet delta in the FAIM data as long as short-term fluctuations (in the gravity wave period range) well after evening twilight are studied (see Figures 6 and 7 in Hannawald et al. (2016)).

Lines 14, 38, 49, 55, 72, 77, 108, 186, 264, 269, 272, 279, and 312: We put commas before and behind "e.g.".

Line 124: "Due to the small FOV of FAIM 3 we renounce the application of a star removal algorithm"

"Please comment briefly on the effect of stars on the results."

Although being rather bright, stars are quite small and have sizes of just a few pixels. Removing them has not altered the spectra in those ranges that were of interest for us, so we refrained from removing them to save computational power and avoid pixel interpolation in the images.

Line 128: "A fitted linear intensity gradient - This fitting is done before the FFT, right?"

Yes, the subtraction of the linear intensity gradient is done before the FFT. We clarified this by adding "...is subtracted from the input images...is applied during the 2d-FFT...".

Line 136: "Wave structures with horizontal wavelengths of half the FOV size still showed a strong bias toward phases 0 or pi"

"Is there a simple reason for this behavior for long wavelengths?"

This behavior is caused by the application of a window function to the image prior to calculating the FFT (a Hann window in our case). The image mean is already subtracted so that wave maxima and minima will be represented by positive and negative values, respectively. The window function will fade the values towards the image borders to zero. If a wave maximum of a pretty large wave is showing only one maximum in the middle of the FOV, the window function will not damp its intensity because the maximum is far away from the image borders. One maximum (in the image center) and two minima (towards the image borders) will be seen in the image. If the phase shifts so that one maximum at each border of the image and one minimum in the image center are visible, the maxima will be damped towards zero by the windowing. What remains is a much lower intensity wave which might be overseen in the spectrum due to its underestimated amplitude. So in conclusion, for large waves (regarding the FOV size), the phase matters

when identifying these waves. Not applying a specific window function (i.e. applying a rectangular window with hard edges at the image borders) is no solution either, as the "sidelobes" of the actual wave signals will have a large effect which would lead to additional pseudo-wave signals in the spectrum with amplitudes almost comparable with the actual wave signal. Our tests showed that all remnants of this effect disappeared for wavelengths below half of the FOV size.

Line 166: "Ca. 63 % of the observed wave events have periods longer than the respective BV period and will be referred to as gravity wave events in the following."

"What about the remaining signatures? What are they? Probably Doppler-shifted GWs?"

Yes, probably Doppler-shifting plays an important role here. As also Referee #1 claimed, setting the BV period as a limit for gravity waves is problematic without having accurate wind data. This is why we refrained from distinguishing between waves above and below the BV period and analysed the wave-like structures as they are. We assume that they are related to instability features like ripples.

We inserted the passage "Please note that we a using the word 'wave' for all wave-like structures we find in the images. The question whether these are actual gravity waves is discussed in section 5." at the beginning of section 4.1 to clarify the more convenient usage of the word 'wave' instead of 'wave-like structures' while presenting the results.

Line 168: "here you distinguish between waves and gravity waves. See my previous comment."

See our previous answer.

Line 184: Referee #2 stated: "Unless the rotational axis is aligned perpendicular to the image plane, the three-dimensional vortex rotation manifests as more than one coherent structure that is moving against or overtaking each other. I read this sentence several times, but didn't really understand it. The grammar (singular/plural) is also not fully correct ("that is moving against or overtaking each other")."

As this sentence belongs to the cylindrical mode, we deleted it entirely.

Line 207: Yes, we were aware of this geometric effect and always read the fastest velocity of each patch to make sure that we did not read its velocity while it was located on a cylinder side.

Line 239: We added the missing degree sign behind "6.2".

Page 8, last paragraph: "If understand correctly, then completely different spatial scales are compared here, right? Several hundred km (spectrometer) vs. a few km (FAIM). Perhaps this can/should be mentioned explicitly."

Yes, it is a good idea to mention this explicitly. We added "Note that the spectrometer GRIPS is sensitive to much larger spatial scales than FAIM." in Line 417.

Line 252: "The directions of propagation are quite uniformly distributed over all quadrants as can be seen in Figure 2."

"Well, looking at the figure, I disagree."

We changed the formulation to "As can be seen in Figure 3, the wave structures we observed exhibit multiple directions."

Line 293: We corrected "In principal" to "In principle".

Line 309: "The here-presented values of K exhibit a magnitude of  $10^{3} - 10^{4}$  m2s-1, which partly agrees with recent results, although some of our values are higher."

"Your values are 2 orders of magnitude larger than the ones published by Lübken. This is quite a large difference. The Liu values are an order of magnitude smaller. Potential reasons should be discussed.

Regarding the Prölss-concept to derive K: I'm not sure, whether this concept applicable to measurements capturing the rotating structures at different stages of the evolution of the structures? The cascade will go from a large eddy to many smaller eddies and I'm not sure what effect is makes, if the structure is analyzed at different times?"

See our changes in this major comment above.

Line 330: "Given that our analyzed episodes are typical representatives of turbulent wave breaking, dynamical heating by gravity wave dissipation would deliver the same effect within few minutes as does chemical heating during an entire day."

"One should keep in mind that the chemical heating is quasi-global and not intermittent in space and time, whereas the dynamical heating is probably quite local. The heating rates only apply to the air volumes affected by the turbulent motion. It would be interesting to estimate what fraction of the global MLT region experiences events (and how many) on a given day."

Yes, we are well aware of the fact that our dynamical heating rates are strongly localized and intermittent, while the chemical heating rates by Marsh are global and referring to the whole day.

We made this clearer to the reader by changing the formulation to "within few minutes at very localized areas in the UMLT as does chemical heating during an entire day for the whole atmosphere."

We agree that it would be quite interesting to rescale the intermittent and localized dynamical heating contributions to a global and continuous reference. Certainly, more than just 25 measurements within one and a half years would be needed to do this properly. We believe that setting up more FAIM3-like systems at various site will be the key for that.

Line 393: "We find an isotropic distribution of directions of propagation"

"Looking at the figures, it is not really isotropic, is it?"

We weakened this statement to "We generally find variable directions of propagation".

Line 347: "and agree mostly with earlier results from rocket and lidar measurements and simulations."

"I disagree. The Lübken values are 2 orders of magnitude smaller (rocket) and the cited lidar values one order of magnitude. Please revise this statement."

We of course revised this statement and changed it to "and are higher than earlier rocket measurements."

Appendix A: "I checked the derivation and it seems to be OK. But I have one general question: The model assumes that the mixing occurs on time scales of one half rotation. However, the rotation will go on and after one full cycle the original state is reached again (assuming a rigid cylinder). And this will go on several more cycles until the vortex disintegrates to smaller vortices. In reality this is of course much more difficult, but I think the model may underestimate the effective mixing time and hence overestimate the turbulent diffusion coefficient. Perhaps this is the reason why your estimates are larger than the other ones?"

We agree to this assumption, see our answer to the major comments above.

"And a minor comment on the appendix: The term "side gas" is quite unusual and I don't know what it means to be honest. Is this a problem with the translation from German? I suggest to use another term."

In the course of our changes from the cylindrical model to the approach of Hecht et al. (2021), we dropped the entire appendix.

**Changes made in the manuscript:**

**Abstract**

Lines 14, 38, 49, 55, 72, 77, 108, 186, 264, 269, 272, 279, and 312: We put commas before and behind "e.g.".

Line 16f: Added "instability features from breaking secondary waves"

Line 17: Replaced "originating from breaking primary waves" by "that were created" and dropped "(westward)" and "(summer)".

Line 20f: We changed "Furthermore, observations of turbulent vortices allowed the estimation of eddy diffusion coefficients in the UMLT from image sequences in 45 cases. Values range around  $10^3 - 10^4 \text{ m}^2 \text{s}^{-1}$  and mostly agree with literature. Turbulently dissipated energy is derived taking into account values of the Brunt-Väisälä frequency based on TIMED-SABER (Thermosphere lonosphere Mesosphere Energetics Dynamics, Sounding of the Atmosphere using Broadband Emission Radiometry) measurements as presented by Wüst et al. (2020). Energy dissipation rates range between 0.63 W kg-1 and 14.21 W kg-1 leading to an approximated maximum heating of 0.2-6.3 K per turbulence event. These are in the same range as the daily chemical heating rates reported by Marsh (2011), which apparently stresses the importance of dynamical energy conversion in the UMLT."

**to**

"We present multiple observations of turbulence episodes captured by our high-resolution airglow imager and estimated the energy dissipation rate in the UMLT from image sequences in 25 cases. Values range around 0.08 and 9.03 W kg-1 and are higher than those in recent literature. The values found here would lead to an approximated localized maximum heating of 0.03-3.02 K per turbulence event. These are in the same range as the daily chemical heating rates for the entire atmosphere reported by Marsh (2011), which apparently stresses the importance of dynamical energy conversion in the UMLT."

**Introduction**

Line 40: We changed "ground" to "troposphere"

Lines 54ff: We changed

"They cause turbulent mixing of the medium, which is described by the eddy diffusion coefficient K. K can be calculated from the eddy radius  $r_e$  and the circumferential velocity  $v_e$  by

$$K = \frac{2}{3\pi} r_e v_e \tag{1}$$

(see e.g. Prölss, 2001. The derivation is outlined in detail in appendix A.). According to Weinstock (1978), knowing K and the BV frequency N, an estimate for the energy dissipation rate  $\epsilon$  - the rate at which turbulent kinetic energy is dissipated into heat at the short-scale end of the energy cascade of the inertial subrange (Li et al., 2016) - can be calculated using

$$K \approx 0.81 \cdot \left(\frac{\epsilon}{N^2}\right).$$
 (2)"

"They cause turbulent mixing of the medium, resulting in the dissipation of turbulent energy at an energy dissipation rate  $\epsilon$ . According to the theory of stratified turbulence,  $\epsilon$  depends on the characteristic length scale L and velocity scale U of the turbulent features. The energy dissipation rate is then given by

$$\epsilon = C_{\epsilon} \frac{U^3}{L}$$
(1)

(see, e.g., Chau et al, 2020; who apply this equation to radar observations of KHIs).  $C_{\epsilon}$  is a constant which is found to be equal to 1 (Gargett, 1999)."

**Lines 70ff: We changed**

"This is why turbulence investigations in the UMLT are challenging and there are only few values of K available at UMLT heights. Lübken et al. (1997) use rocket measurements to retrieve K and  $\epsilon$  in the height range 65-120 km. Liu (2009) presents a method for the estimation of K from gravity wave momentum fluxes derived from lidar data. Baumgarten & Fritts (2014) use imaging techniques of mesospheric noctilucent clouds to investigate the formation of KHIs and the onset of turbulence."

**to**

"This is why turbulence investigations in the UMLT are challenging and there are only few values of  $\epsilon$  available at UMLT heights. Lübken et al. (1997) use rocket measurements to retrieve  $\epsilon$  in the height range 65-120 km. Baumgarten & Fritts (2014) use imaging techniques of mesospheric noctilucent clouds to investigate the formation of KHIs and the onset of turbulence.".

Line 76: We changed "These include..." to "Remote sensing techniques include..."

Line 83: We changed "Proceedings" to "Improvements"

**Data Basis**

Line 138: We inserted "...from the input images..."during the 2d-FFT".

Line 156: We inserted "An exemplary event and the respective 2-dimensional spectrum are shown in Figure 1. We often observe episodes of turbulence in our image series that exhibit the typical dynamics of vortex formation and quasi-chaotic behavior."

Line 164: Replaced "45" by "25" and "vortex" by "turbulence"

Lines 166ff: Omitted "For both, gravity wave statistics (section 4.1) and the calculation of the energy dissipation rate (section 4.2) the BV frequency is required, which is adapted from the climatology presented by Wüst et al. (2020). It is based on TIMED-SABER (Thermosphere Ionosphere Mesosphere Energetics Dynamics, Sounding of the Atmosphere using Broadband Emission Radiometry) temperature data and takes into account the seasonal variability of the angular BV frequency. The climatology of the grid point (45° N, 10° E) is used, which is closest to our FOV. Depending on the day of the year (DoY) the BV frequency is given by

 $N = 2.20 \cdot 10^{-2} \mathrm{s}^{-2} + 0.19 \cdot 10^{-2} \mathrm{s}^{-2} \sin\left(\frac{2\pi}{324.51\mathrm{d}} \cdot DoY - 2.02\right) + 0.05 \cdot 10^{-2} \mathrm{s}^{-2} \sin\left(\frac{2\pi}{180.00\mathrm{d}} \cdot DoY + 1.61\right).$  (3)

to

We use an uncertainty of ±5% as according to Wüst et al. (2020) 91% of their data lie within this range around the harmonic approximation. The BV period is then referred to as  $\tau_{BV} = \frac{2\pi}{N}$ ."

**Results**

We inserted the passage "Please note that we a using the word 'wave' for all wave-like structures we find in the images. The question whether these are actual gravity waves is discussed in section 5." at the beginning of section 4.1.

Line 165 ff: We dropped "For each wave event the individual BV period is calculated based on Eq. (3) and the DoY. Ca. 63% of the observed wave events have a period longer than the respective BV period and will be referred to as gravity wave events in the following. Their statistical contribution is highlighted in grey in Fehler! Verweisquelle konnte nicht gefunden werden.."

Line 168 f: We dropped "The median value of gravity wave periods is found at 517 s (8.6 min).".

Line 169: 17.6 m/s changed to 139.8 m/s; 7.9 m/s changed to 13.3 m/s

Line 170: 3.3 m/s changed to 10.3 m/s. 50.7 % (49.3 %) changed to 52.7 % (47.3 %). Dropped the word "gravity".

Line 171: Dropped the word "gravity".

Line 172: 5.4 m/s (5.3 m/s) changed to 9.4 m/s (8.2 m/s). 3.0 m/s (3.1 m/s) changed to 8.9 m/s (7.4 m/s).

Line 173: 53.8 % changed to 56.0 %. 46.2 % changed to 44.0 %.

Line 174: 47.9 % changed to 49.5 %.

Line 175: 52.1 % changed to 50.5 %. 44.4 % (53.5 %) changed to 42.0 % (55.7 %).

Line 176: 4.7 m/s (5.3 m/s) changed to 7.5 m/s (9.4 m/s).

Line 177: 3.0 m/s (3.2 m/s) changed to 7.1 m/s (8.7 m/s).

Lines 192 ff: We added: "To give an impression of the turbulent dynamics we observe, we present four of our turbulence episodes as video supplement. On 16 November 2017, 02:16 UTC, the turbulent breakdown of parts of an extended wave field can be observed (video 1). On 6 December 2017, 00:26 UTC, several fronts seem to be building up and form rotating vortices (video 2). This can be observed even clearer on 14 October 2018, 17:08 UTC, where the residual movement of turbulent features can be well recognized above the general background movement (video 3). On 4 November 2018, 19:18 UTC, breaking wave fronts seem to form rotating structures of nearly cylindrical shape, while these are accompanied by other turbulently moving eddies (video 4).

[revised manuscript text omitted]

**The values of K range from 0.12 to $1.94 \cdot 10^4 \text{ m}^2 \text{s}^{-1}$ . The mean value is $0.76 \cdot 10^4 \text{ m}^2 \text{s}^{-1}$ "**

to

"As stated in section 3 we found 25 episodes of turbulence that allowed the derivation of L and  $v_{res}$ . Using equation (1), the energy dissipation rate is then calculated by  $\epsilon = \frac{v_{res}^3}{L}$ . The resulting values are shown in **Fehler! Verweisquelle konnte nicht gefunden werden.** We assume a general read-out error of  $\pm 3$  pixels, which corresponds to a distance of  $\pm 72$  m. Velocities are determined by reading the distance a feature covers within an episode of at least ten images, which corresponds to a time span of 28 s. Thus, velocities are estimated with an error of  $\pm 2.6 \text{ ms}^{-1}$ . The arising uncertainties of  $\epsilon$  are calculated following the rules of error propagation.

The values of  $\epsilon {\rm range}$  from 0.08 to 9.03  $\,$  W kg^{-1}. The median value is 1.45 W kg''

**Lines 240ff: We dropped**

"(standard deviation of  $0.53 \cdot 10^4 \text{ m}^2 \text{s}^{-1}$ ) and we retrieve a median of  $0.59 \cdot 10^4 \text{ m}^2 \text{s}^{-1}$ . It is difficult to exactly quantify the error of manual parameter determination from the images. However, in this work we rather focus on the order of magnitude of *K*. When calculating *K* two distance values are read from the images (one for the vortex size and one for the determination of the circumferential speed). Considering Eq. (1), a mistake of factor 10 is made for *K* if these distances are misread by a factor of at least  $\sqrt{10}$ . The shortest (and therefore most difficult to determine) diameter in our analysed examples was 768 m. For an error of one order of magnitude of *K* this distance must be misread as either shorter than 243 m or longer than 2428 m, i.e. a distance of 32 pixels must be wrongly interpreted as shorter than 9 pixels or longer than 101 pixels. This lies far beyond the readout uncertainty of  $\pm 3$  pixels we introduced above and can be assumed to be much worse than any read-out error one would normally make.

The energy dissipation rate  $\epsilon$  can be estimated from the eddy diffusion coefficient K according to Eq. (2) using the BV frequency as described by Eq. (3).

As can be seen in **Fehler! Verweisquelle konnte nicht gefunden werden.** the energy dissipation rate of the observed turbulence events is in the range  $0.63 - 14.21 W kg^{-1}$ ."

Line 254: We updated "146 s" to "241 s" and "2.4" to "4.0".

Line 258: We updated "220" to "30" and "6346" to "3015".

Line 262: We updated "0.2-6.3 K" to "0.03-3.02 K" and dropped "64% (21 out of 33) of these values are larger than one Kelvin."

Line 268: We inserted a missing "°"

Lines 277ff: We changed "We find a slight but significant anticorrelation for gravity wave periods in the range 122-207 min. For these periods the mean value of the correlation coefficient is -0.46. The highest coefficient of anticorrelation is -0.52 at a period of 178 min."

"We find almost no significant correlation for any gravity wave period."

**Discussion:**

Line 280: We replaced "The directions of propagation are quite uniformly distributed over all quadrants as can be seen in Figure 2" by "As can be seen in Figure 3, the wave structures we observed exhibit multiple directions."

Line 288: We replaced "(positive zonal phase speed) and in westward direction during summer. Although this tendency is quite weak," by "whereas zonal directions are quite balanced during summer. Although the eastward tendency during winter is quite weak,".

Line 290: We added "tropospheric and".

Lines 292ff: We replaced "The reversed stratospheric winds during summer would consequently allow some more eastward travelling gravity waves to propagate upward (see e.g. Hoffmann et al., 2010; Hannawald et al., 2019). Since the highest observed phase speed of waves with periods longer than the BV period is only 17.6 m s-1, it can be assumed that in the majority we do not observe gravity waves that are originating from low altitudes and are fast enough not to be blocked by the stratospheric wind fields."

**by**

"During summer the stratospheric winds reverse to westward direction, so that eastward oriented gravity waves are filtered in the tropopause and westward oriented gravity waves are filtered in the stratosphere (see, e.g., Hoffmann et al., 2010; Hannawald et al., 2019).".

**We inserted**

"As we have no accompanying wind measurements in the height of our observations it is difficult to decide by means of the period whether the wave structures presented in section 4.1 are small-scale gravity waves or instability features. Ca. 63 % of the observed wave events have an observed period above the BV period (here we used the climatology presented by Wüst et al., 2020), however these could also be Doppler-shifted instability features instead of gravity waves. While the distinction between largely extended wave-fields (bands) and small localized wave structures that are related to instability (ripples) is often made at a horizontal wavelength of 10 - 20 km (Taylor et al., 1997; Nakamura et al., 1999), Li et al. (2017) remark that even structures with horizontal wavelengths of 5 - 10 km may sometimes be gravity waves rather than instability features."

after this passage.

Line 302 ff: We changed "Considering the directional distribution, it is possible that the major part of our waves may" to "If this would be true for our small-scale wave structures, they might rather".

Line 321 f: We changed "southward in 62 % of cases." to "southward in 71 % of cases.".

We inserted

to

"However, regarding the small horizontal wavelengths below 4.5 km, it is more likely that the major part of the observations presented in section 4.1 are related to instability features. The quite slow phase speeds (mean value 13.3 m / ) are one hint for this as typical gravity wave phase speeds accumulate around 40 m / s (see, e.g., Wachter et al., 2015 and Wüst et al., 2018). If Figure 3b was the phase speed distribution of gravity waves, it is likely that a majority of them would encounter critical levels somewhere and would not be observable in the OH\* layer."

**after this passage.**

Line 371 ff: We replaced "Li et al. (2017) report that ripples are hard to distinguish from small-scale gravity waves. Height-resolved measurements of the horizontal wind would be needed to determine the local wind shear and make a profound statement about atmospheric instability. Nevertheless,"

**by**

"Considering the fact that the directional peculiarities of our observed wave events fit well with the expected behavior of secondary gravity waves, as discussed above, support the scenario of the wave structures being ripples from dynamic instabilities of secondary gravity waves, that originate from the stratospheric and mesospheric jet.".

**We added**

"Nevertheless, height-resolved measurements of the horizontal wind would be needed to determine the local wind shear and make a profound statement about atmospheric instability. It has to be kept in mind that a 2d-FFT was used. Thus, periodic structured are assumed to be stationary, i.e., they extend over the entire image. Faint structures that appear only in small parts of the image (as does for example the 550m wave packet in Sedlak et al., 2016; Fig. 2) would be underrepresented by this analysis."

at the end of the discussion section of the wave statistics.

**Lines 384ff:**

**We changed**

"There are still very few measurements of turbulent eddy diffusion coefficients in the UMLT. Lübken (1997) reports K to be around  $10^1 - 10^2 \text{ m}^2 \text{s}^{-1}$  at a height of 87 km at high latitudes. Hodges (1969) states that the eddy diffusion coefficient caused by gravity waves is typically around  $10^3 \text{ m}^2 \text{s}^{-1}$ . According to the CIRA (Committee on Space Research (COSPAR) International Reference Atmosphere) climatology of 1986 (NASA National Space Science Data Center, 2007) global values range between magnitudes of  $10^2$  and  $10^3 \text{ m}^2 \text{s}^{-1}$ . LIDAR measurements above New Mexico, USA deliver values that vary strongly around a magnitude of  $10^2 \text{ m}^2 \text{s}^{-1}$  (Liu, 2009). Smith (2012) notes that the WACCM (Whole Atmosphere Community Climate Model) climatology exhibits rather small values with magnitude  $10^1 \text{ m}^2 \text{s}^{-1}$  and that the huge discrepancies of K estimates cannot be fully explained yet. The here-presented values of K exhibit a magnitude of  $10^3 - 10^4 \text{ m}^2 \text{s}^{-1}$ , which partly agrees with recent results, although some of our values are higher."

"Measuring the eddy diffusion coefficient in the UMLT is still challenging and there are only few studies yet. Rocket measurements of Lübken (1997) deliver energy dissipation rates between ca. 0.01 and 0.1W kg-1 between 85 and 90 km height at high latitudes. Chau et al. (2020) find an energy dissipation rate of 1.125 W kg-1 for their KHI event observed in the summer mesopause and state that this a rather high value compared to the findings of Lübken et al. (2002). Hocking (1999) provides a rescaled overview of earlier values of the energy dissipation rate and these have a maximum magnitude of 0.1 W kg-1. Hecht et al. (2021) derive a value of 0.97 W kg-1 from airglow images of a KHI event. Ranging from 0.08 up to 9.03 W kg-1 the values of energy dissipation rate derived here are higher than reported by other studies. However, the median value of 1.45 W kg-1 is not too far away from the values of Chau et al. (2020) and Hecht et al. (2021)."

Lines 402f: We added "– except for the studies of Hecht et al. (2021), whose value is quite similar to the median value of our data -,"

Lines 405ff: We dropped "Nevertheless, the agreement of the above-mentioned authors on eddy parameters in the UMLT is quite good, considering the fact that energy dissipation rate in the upper troposphere and lower stratosphere varies by a factor of more than five orders of magnitude (Li et al., 2016).".

Line 408: Changed "vortex" to "turbulence"

Line 409: Changed "Circumferential speed and vortex radius" to "The length scale and velocity scale of turbulent features"

Line 410: "and we" changed to ". We"

Lines 412f: Inserted "However, using equation (1) velocity dominates the length scale due to its power of 3, so that  $\epsilon$  strongly depends on a parameter, which is quite difficult to extract from the images."

Line 414: Dropped "eddy diffusion coefficient and"

Line 415: Replaced "eddy diffusion coefficients" by "energy dissipation rate" and replaced "significant anticorrelation" by "no significant correlation"

Lines 416: Changed "122-207 min" to "6-480 min"

Lines 416ff: Changed

[revised manuscript text omitted]

**Appendix**

We dropped Appendix A.

**Figures**

Figure 1 is now a sample event with its 2D-FFT spectrum.

Figure 2 (former Figure 1): Histograms of wave parameters without separation by the BV period

Figure 3 is now former Figure 2

Figure 4 (former Figure 3): New snapshot showing how to read the new wave parameters from the images.

Former Figure 4: dropped

Figure 5: Plot of temporal course and histogram of  $\epsilon$  (no K values anymore)

Figure 6: Updated histogram of temperature changes

Figure 7: Updated plot of correlation analysis

Figure 8 (concept of rotating cylinder): dropped

**Tables**

We updated Table 1 by dropping those lines referring to events which we could not analyse anymore with the new method. We dropped the columns "DoY", "K" and "Angular BV Frequency".

**Gravity wave instability structures and turbulence from more than one and a half years of OH\* airglow imager observations in Slovenia**

René Sedlak1, Patrick Hannawald1,2, Carsten Schmidt2, Sabine Wüst2, Michael Bittner1,2, and Samo Stanič3

1Institute of Physics, University of Augsburg, Augsburg, Germany

[revised manuscript text omitted]
 10^{-2} \mathrm{s}^{-2} + 0.19 \cdot 10^{-2} \mathrm{s}^{-2} \sin\left(\frac{2\pi}{324.51d} \cdot DoY - 2.02\right) + 0.05 \cdot 10^{-2} \mathrm{s}^{-2} \sin\left(\frac{2\pi}{180.00d} \cdot DoY + 1.61\right).$$
(3)

We use an uncertainty of  $\pm 5\%$  as according to Wüst et al. (2020) 91% of their data lie within this range around the harmonic approximation. The BV period is then referred to as  $\tau_{BV} = \frac{2\pi}{N}$ .

**175 **4 Results**

**4.1 Statistics of Wave Parameters**

The wave statistics are presented in Figure 2Figure 1. Please note that we a using the word 'wave' for all wave-like structures we find in the images. The question whether these are actual gravity waves is discussed in section 5. For each wave event the individual BV period is calculated based on Eq. (3) and the DoY. Ca. 63% of the observed wave events have a period longer

**180 than the respective BV period and will be referred to as gravity wave events in the following. Their statistical contribution is highlighted in grey in Figure 1.**

Wave periods range from 21 s to 1498 s (25 min). The median wave period is 359 s (6 min). The median value of gravity wave periods is found at 517 s (8.6 min). The maximum phase speed is  $\frac{17.6139.8}{50.752.7}$  m s-1 with an average value of  $\frac{7.913.3}{50.752.7}$  m s-1 and a standard deviation of  $\frac{310}{3}.3 \text{ m s}^{-1}$ . As concerns the zonal distribution,  $\frac{50.752.7}{50.752.7}$ % ( $\frac{49.347.3}{50.752.7}$ %) 
[revised manuscript text omitted]

---

## Author Response (AR2)

The authors would like to thank the two anonymous referees for their positive reviews and Referee #2 for the additional comments, which will be answered in the following.

Line 140: Zeropadding: First, the image is padded to a size consisting solely of prime factors 2,3, and 5 (requirement of the used variant of the Fast Fourier Transform) which in our case is 432 x 432 Pixels. It is then additionally padded with zeros for a finer resolution of the spectrum peaks. The number of zeros is twice the number of image pixels in each direction. This leads to a final image which is of size 2160x2160 Pixels (2*432*2 + 1*432).

> We inserted the relevant information in the manuscript by adding „(to a size of 2160 x 2160 pixels)" in line 128.

Fig 1: The axes in the 2d spectrum display values of the inverse wavelength (1/lambda, not 2pi/lambda). To avoid confusion, we adjusted the axis titles from k to $\lambda^{-1}$ and dropped the term 'in the k-space' in the caption.

Line 265: As Referee #2 assumes correctly, the analysis of the eddy diffusion coefficient is a remnant of the former paper version. Indeed, in the revised version, we analyzed the correlation between gravity wave activity and the energy dissipation rate $\epsilon$. Unfortunately, we forgot to update the corresponding text passages.

> We corrected "eddy diffusion coefficient(s)" to "energy dissipation rate(s)" in lines 201, 212, 213, 280 and in the caption of Fig. 7.

Figure 7: All in all, 475 correlation coefficients have been calculated. We correlated each of the SWI time series (one for each gravity wave period 6-480 min; 1 min steps) with the time series of energy dissipation rate. The latter one has no period-dependence. We hope that we have made this clearer by rephrasing Lines 211 ff. to "The time series of nocturnal SWI is restricted to those nights that exhibited at least one of the turbulence episodes presented above. For each gravity wave period between 6 and 480 min (1 min steps), the correlation between the SWI at the respective period and the energy dissipation rate has been calculated.".

> As Referee #2 suggested, we now show exemplarily the underlying time series that correspond to the most significant positive correlation of energy dissipation rate and gravity wave activity in the long-period range in Figure 7b. The caption has been expanded by "b) Comparison of $\epsilon$ and the SWI at period 401 min, which is closest to a significant positive correlation in the long-periodic part of the gravity wave spectrum (Pearson Correlation Coefficient 0.45).".

Line 298: We do mean "above the BV period" (below the BV frequency). As in the first review Referee #1 stated correctly, the small-scale features we observe are more likely instability features (like ripples) rather than gravity waves. If this was true, periods above the BV period (apparently in the gravity wave regime) could be explained by assuming Doppler-shifted instability features. Periods below the BV period could accordingly be related to high-frequency instability features, that are not subject to Doppler-shifting (and do not appear in the gravity wave period range to the steady observer).

Line 382: We corrected "periodic structured" to "periodic structures".

Line 416: See our comment on Line 265 – the term "energy dissipation rate" is the correct one. As Referee #2 states correctly, there is a slight positive correlation of energy dissipation rate and gravity wave activity with periods > 400 min, which is close to significant. We now mention this in line 216: "Long-periodic SWI (periods > 400 min) shows a slight positive correlation with the energy dissipation rate, which is nearly significant." and provide a very careful interpretation in lines 304 ff:

- Replaced "cannot" by "can hardly"
- Added "The slight positive correlation with gravity wave activity at periods larger than 400 min may point to a special contribution of long-period gravity waves to the turbulence events we observe. However, this remains speculative at the current stage of research, since this correlation is beyond the level of significance."

Equation 1, Figure 4 and line 205 and the following: Referee #2 wonders whether the velocity of the eddy relative to the background motion is the correct quantity to use as U in equation 1 or if the eddy circumferential velocity should rather be used instead. Having read the considerations of Hecht et al. (2021), Referee #1 recommended in the first review, we believe that the eddy velocity relative to the background, referred to as root-mean-square velocity, might be the better choice here, as our data are quite comparable to those of Hecht and we agree to his argumentation. In our opinion, the circumferential velocity is the quantity to use when applying a rotating cylinder model, as we did in the original version of this manuscript. This quite idealistic model, however, is accompanied by many uncertainties, as Referee #1 discussed in great detail in his first review. Using the circumferential speed along with the rotating cylinder model might provide an upper limit of dissipated energy in an idealized twin-scenario of the observed episode. As the focus of this work is on the presentation and analysis of observational data, we decided that using the root-mean-square approach applied by Hecht et al. (2021) may be the more realistic approach in our case.